# Neuromorphic intermediate representation: A unified instruction set for interoperable brain-inspired computing

Jens E. Pedersen [1,15] ✉, Steven Abreu [2,3,15], Matthias Jobst [4,5], Gregor Lenz[6], Vittorio Fra [7], Felix Christian Bauer [8], Dylan Richard Muir [8], Peng Zhou[9], Bernhard Vogginger [4], Kade Heckel [10], Gianvito Urgese [7], Sadasivan Shankar [11,12], Terrence C. Stewart[13], Sadique Sheik[8] & Jason K. Eshraghian[14]

Spiking neural networks and neuromorphic hardware platforms that simulate neuronal dynamics are getting wide attention and are being applied to many relevant problems using Machine Learning. Despite a well-established mathematical foundation for neural dynamics, there exists numerous software and hardware solutions and stacks whose variability makes it difficult to reproduce findings. Here, we establish a common reference frame for computations in digital neuromorphic systems, titled Neuromorphic Intermediate Representation (NIR). NIR defines a set of computational and composable model primitives as hybrid systems combining continuous-time dynamics and discrete events. By abstracting away assumptions around discretization and hardware constraints, NIR faithfully captures the computational model, while bridging differences between the evaluated implementation and the underlying mathematical formalism. NIR supports an unprecedented number of neuromorphic systems, which we demonstrate by reproducing three spiking neural network models of different complexity across 7 neuromorphic simulators and 4 digital hardware platforms. NIR decouples the development of neuromorphic hardware and software, enabling interoperability between platforms and improving accessibility to multiple neuromorphic technologies. We believe that NIR is a key next step in brain-inspired hardware-software co-evolution, enabling research towards the implementation of energy efficient computational principles of nervous systems. NIR is available at neuroir.org

The human brain is an exemplar of computational efficiency, out-performing today's advanced machine learning models and hardware in multiple measures, particularly regarding energy consumption. State-of-the-art machine learning algorithms require vast amounts of energy and computational resources, especially when trained on large datasets[1,2], whereas the brain operates on a tiny power budget, learning and performing a myriad of complex tasks seamlessly. Unlike digital computations that rely on precise binary logic, the brain's computations have a significant analog component in nature, utilizing gradients of ion concentrations and action potentials, which can be represented as unary activations, commonly referred to as spikes. Neuromorphic computing, defined as "circuits that emulate the temporal processing of signals in the brain" [ref. 3, p. 361], draws inspiration from the efficiency of the nervous systems to rethink the principles of information

processing. As such, neuromorphic computing stands in contrast to the conventional and widespread von Neumann architecture and relies on entirely new algorithms, software tools, and hardware[4].

Much like how the brain can be thought of as a physical embodiment of the neural code consisting of architecture, hardware, and software, contemporary neuromorphic hardware and software are often developed together[3]. The blurring of top-down and bottom-up approaches makes it difficult to extrapolate clear abstractions that generalize to other technology stacks, which impedes incremental technological progress[5]. The blurring of top-down and bottom-up approaches makes it difficult to extrapolate clear abstractions that generalize to other technology stacks, which impedes incremental technological progress. As a result, we face a landscape of heterogeneous neuromorphic simulators and design tools. By identifying domain-specific computational primitives, we argue that a layer of abstraction can be established to achieve greater interoperability and, in turn: (1) allow hardware and software efforts to develop independently and more rapidly, (2) lower the barrier-of-entry for newcomers, (3) provide theoreticians with a stable representation to study fundamentals of computing across different neuromorphic circuits, and (4) possibly enable more energy-efficient computing.

The field of deep learning faced similar heterogeneity, and incompatibility challenges a decade ago, which have been addressed by intermediate representations (IR) and compiler frameworks such as ONNX[6], MLIR[7], XLA[8], and TVM[9]. Note that, by definition, IRs serve exclusively as a model description, while the compilers carry out the translation between platforms, although they are often mixed in practice. These tools have reduced the gap between the simulation of an application and different hardware accelerators, enabling models written in higher-level languages to be mapped to different classes of hardware backends, such as CPUs, GPUs, TPUs, and FPGAs. Notably, the development of sophisticated compiler frameworks was only possible once a stable representation and exchange format was established. Following this precedent, NIR aims to establish a stable representation and exchange format for neuromorphic computing, as a foundation on which more specialized compiler frameworks can be built. Similarly, cross-framework compatibility for deep learning has been addressed by approaches such as Ivy[10] or MMdnn[11]. Given the nature of clocked computation in the traditional computing stack, deep learning representations are based on digital instructions in discrete time. This contrasts our definition of neuromorphic models that are best captured as continuous-time dynamical systems. Any representation of a continuous system as a sequence of digital instructions implies using numerical integration techniques, possibly introducing numerical errors during the computation of the desired dynamics. In addition, neuromorphic simulators and hardware platforms operate on the level of neuronal dynamics, a fundamentally different abstraction than digital intermediate representations for conventional machine instruction set-based architectures. This renders existing intermediate representations such as ONNX and MLIR suboptimal for describing neuromorphic computations.

Numerous configurable neuromorphic systems have been developed over the last decades[12]. The implementation approaches range from analog / mixed-signal[13–16], over digital[17–19], hybrid analog/digital[20] to processor-based[21,22] solutions. We refer to Furber 2016[23] and Thakur et al.[24] for a detailed overview and comparison of the most prominent large-scale systems. Most of the systems offer custom interfaces for programming the chips, such as the Corelet approach[25] for TrueNorth[17] or PyNCS[26] for the UZH|ETHZ chips[20].

Several works emphasize the need to make neuromorphic hardware and simulators more user-friendly and accessible[4,27,28] through co-design[5] and shared representations[29]. PyNN[30] is a Python library that allows users to define spiking neural network (SNN) models in a simulator-independent language. PyNN models can be run without modifications on different neural simulators (NEST[31], Neuron[32],

Arbor[33], Brian2[34], CARLSim[35]). PyNN has developed into the most widespread common interface to neuromorphic hardware, supporting the Heidelberg Spikey chip[36], the BrainScaleS systems[37], and SpiNNaker1[38]. NeuroML[39] provides a serializable set of biological cell and network models which emphasizes computational correctness for neuroscientific studies. The popular Brian2 simulator[34] has been extended to interface with other tools, such as the SNN-GPU simulator GeNN[40], or to emulate the Intel Loihi chip[41].

Several frameworks have been developed to design neuromorphic algorithms. Fugu[42] provides a high-level API to define and combine spiking neural algorithms. It generates an SNN graph as an intermediate representation, which can serve as input to neuromorphic simulators or hardware compilers. Similarly, Lava[43] is a framework for developing brain-inspired neural network models and mapping them to Intel's digital neuromorphic hardware. Its representation is based on the Communicating Sequential Processes (CSP) principle[44], where neurons are represented by processes that send spikes via channels connected to other processes. Lava significantly simplifies the programming of the neuromorphic chip Loihi 2[45]. However, at its core, it requires synchronization messages for triggering neuron updates, which is incompatible with applications exhibiting continuous-time behaviors. Nengo[46] is a mature library for brain simulation and deep learning-inspired spiking networks. Similar to PyNN, it has been connected to several neuromorphic systems such as BrainDrop[15], Loihi, SpiNNaker1, and FPGAs. In addition to Fugu, Lava, and Nengo, there are frameworks such as the SNN-Toolbox[47], NxTF[48], or hxtorch.snn[49] for converting neural network models from ML frameworks to specific SNN simulators and neuromorphic hardware. Zhang et al.[50] have developed an abstraction hierarchy for brain-inspired computing with loose guarantees, portable across both von Neumann and neuromorphic architectures. Other interfaces focus on particular computing elements, such as crossbar arrays using beyond-CMOS technologies[29,51], or ReLU and leaky integrate-and-fire units[52]. Unfortunately, none of these standards have spread beyond their intended subdomain within neuromorphic computing.

Theoretically, our work is inspired by the work on signal flow graphs[53], linear time-invariant systems[54], and mealy machines[55] for the description of composable circuit components with applications to digital, or analog, or hybrid systems. Compiler infrastructures projects like LLVM[56], and later MLIR[7], has been leading the field in optimizing and transforming code for a wide range of programs and architectures.

In this work, we (1) derive and implement a set of model-centric computational primitives that are common to multiple neuromorphic software and hardware systems, (2) illustrate the application of NIR across 7 different simulators and 4 different hardware platforms, (3) demonstrate flexible cross-platform deployment of NIR models, and (4) evaluate exemplary neuromorphic models defined in NIR across all 11 supported platforms—in hardware and software. This work comes at a decisive moment for novel computing technologies, where the power consumption of neural networks is growing at unprecedented rates, and when neuromorphic platforms are available at scale to the consumer for the first time.

## Results

In a collaboration between academia and industry, we propose and illustrate the Neuromorphic Intermediate Representation: a model-centric abstraction layer that simplifies the translation between simulated applications, neuromorphic software, and digital hardware platforms. NIR currently links 7 neuromorphic simulators (Lava[43], Nengo[46], Norse[57], Rockpool[58], Sinabs[59], snnTorch[60], and Spyx[61]) and 4 digital neuromorphic hardware platforms (Loihi 2[62] via Lava, Speck via Sinabs, SpiNNaker2, and Xylo[63] via Rockpool). NIR represents computations as graphs, where each node represents a computational primitive defined by a hybrid continuous-time dynamical system. This idealized description provides three distinct advantages: (1) it avoids

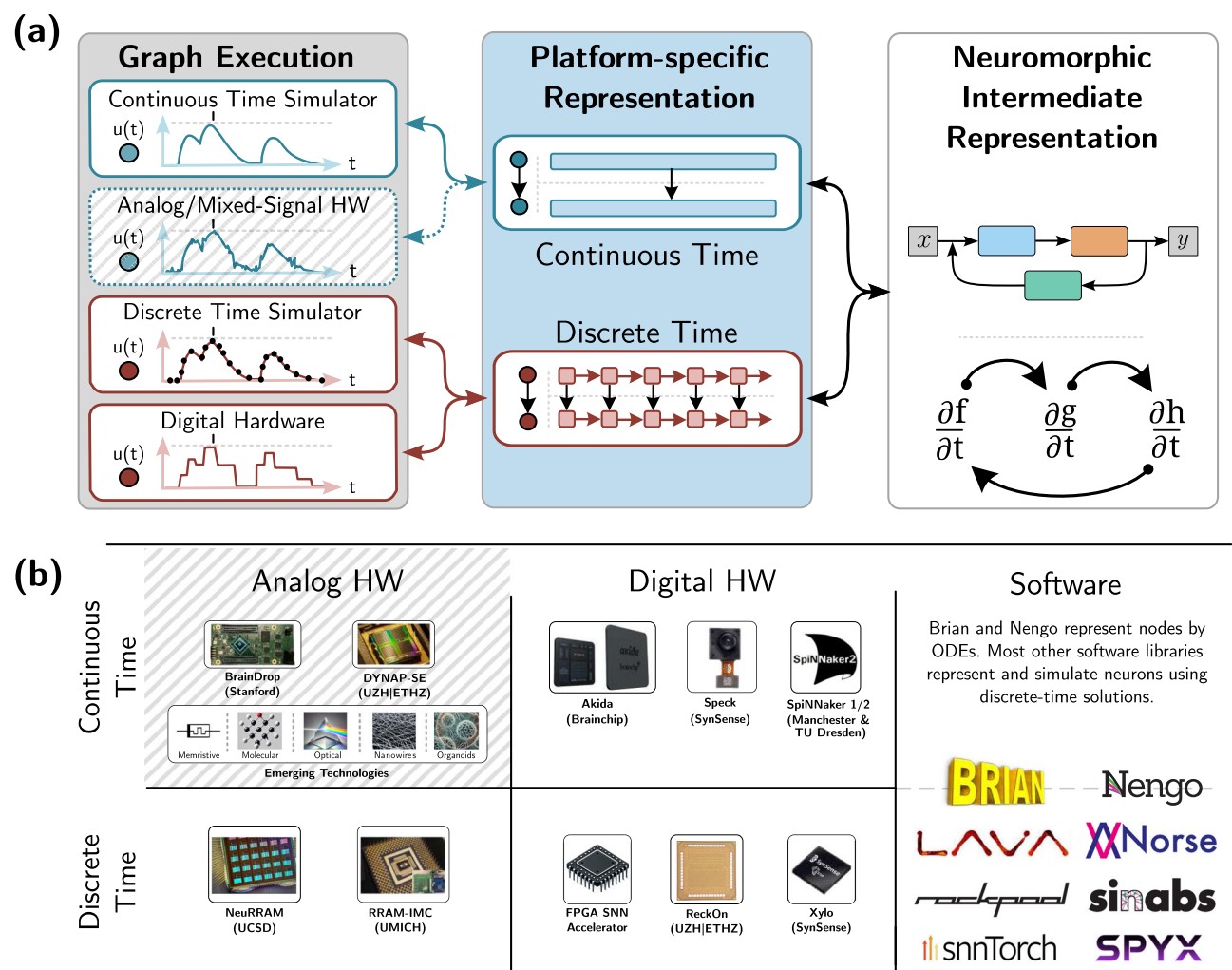

**Fig. 1 | High-level overview of NIR. a** NIR allows for continuous-time representation of specific models that can then be executed on continuous-time hardware or simulators or discretized for use on discrete-time hardware or simulators. **b** A taxonomy of discrete and continuous time hardware and simulators. Some representative hardware systems are shown. Analog Continuous Time: BrainDrop[15], DYNAP-SE[20], and emerging technologies. These systems are grayed out because they are not covered in the present paper. Analog Discrete Time: NeuRRAM[97], UMich RRAM-IMC[98]. Digital Continuous Time: Akida, SpiNNaker 1/2[21,22], Speck. Note that we include chips with asynchronous routing in this category. Digital Discrete Time: Generic FPGA-based SNN Accelerators, ReckOn[99], Xylo[100].

any assumptions around discretization or hardware constraints, (2) it provides a reference model against which implementations can be compared, and (3) it decouples the software description from the hardware layer, allowing seamless integration with mixed-signal chips, as well as analog hardware, and hybrid digital-analog systems. The computational graph representation is shown in Fig. 1, along with its relation to digital and analog hardware as well as software, independently of the underlying execution model being continuous-time or discrete-time. While the continuous nature of NIR can be applied to mixed-signal hardware platforms, this work focuses on digital simulators and hardware platforms. It is also important to point out that the conversion between continuous-time dynamical systems and digital platforms will give divergent results. We will further define and measure these discrepancies in later sections.

NIR defines a set of primitives as coupled hybrid systems, formally introduced in the Methods Section, that can be arbitrarily composed, as shown in Figs. 1 and 2. NIR operates on a predefined set of computational primitives rather than custom neural equations, which cannot easily be supported on efficient and specialized hardware. This approach is motivated by the nearly universal agreement on a few computational models, such as the leaky integrator and the integrate-and-fire models, in prevalent neuromorphic hardware platforms and

simulators. It also allows us to achieve strong generality early on; NIR is presently supported by four digital neuromorphic hardware platforms: Intel Loihi 2[43], SynSense Speck, SpiNNaker2, and SynSense Xylo[63]. In addition, 7 software simulators now support NIR: Lava (and Lava-DL)[43], Nengo[46], Norse[57], Rockpool[58], Sinabs[59], snnTorch[60], and Spyx[61].

The primary strength of NIR resides in its ability to reproduce the same computation on heterogeneous platforms, abstracting away platform-specific differences. Thus, NIR is able to translate a computational model into multiple and possibly different, hardware platforms; training or defining models can be done independent of the substrate carrying out the computation. Below, in Section Experiments, we demonstrate how an NIR graph can be executed and supported by the above-mentioned platforms for three different tasks: a leaky integrate-and-fire model, a spiking convolutional network, and a spiking recurrent neural network. Later, in Sections NIR axioms and Computational primitives, we establish the axioms underpinning NIR graphs and list the computational primitives we presently support.

## Experiments
We conduct a series of experiments spanning diverse neuromorphic platforms to gauge the robustness and versatility of NIR. We chose

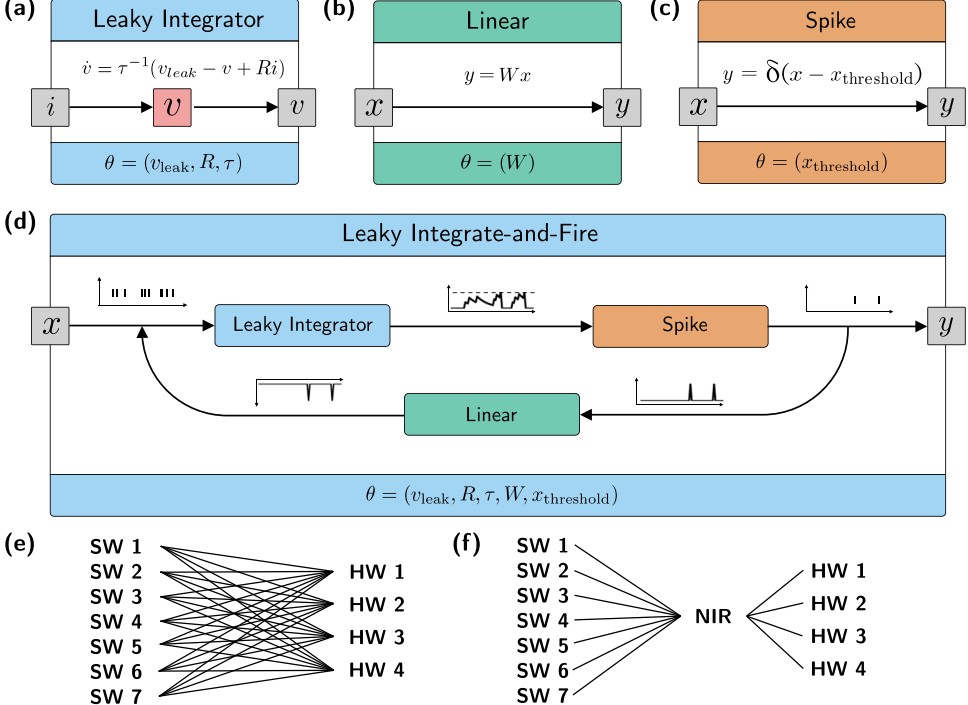

**Fig. 2 | Composition of primitives and their mapping to and from software and hardware systems. a–d** shows four NIR primitives, where the name of the primitive is highlighted on top, the implemented computation is illustrated in the white box, and the parameters that are stored with the NIR graph are highlighted on the bottom. **a** shows a stateful primitive, while (**b**) and (**c**) show stateless primitives, and (**d**) show a higher-order primitive. **e–f** illustrate the concept of an intermediate representation (IR) with 7 software (SW) and 4 hardware (HW) backends. Instead of 30 different compilers covering all $m \times n$ cases (**e**), only $m + n$ interfaces between each hardware and software platform to NIR are necessary (**f**).

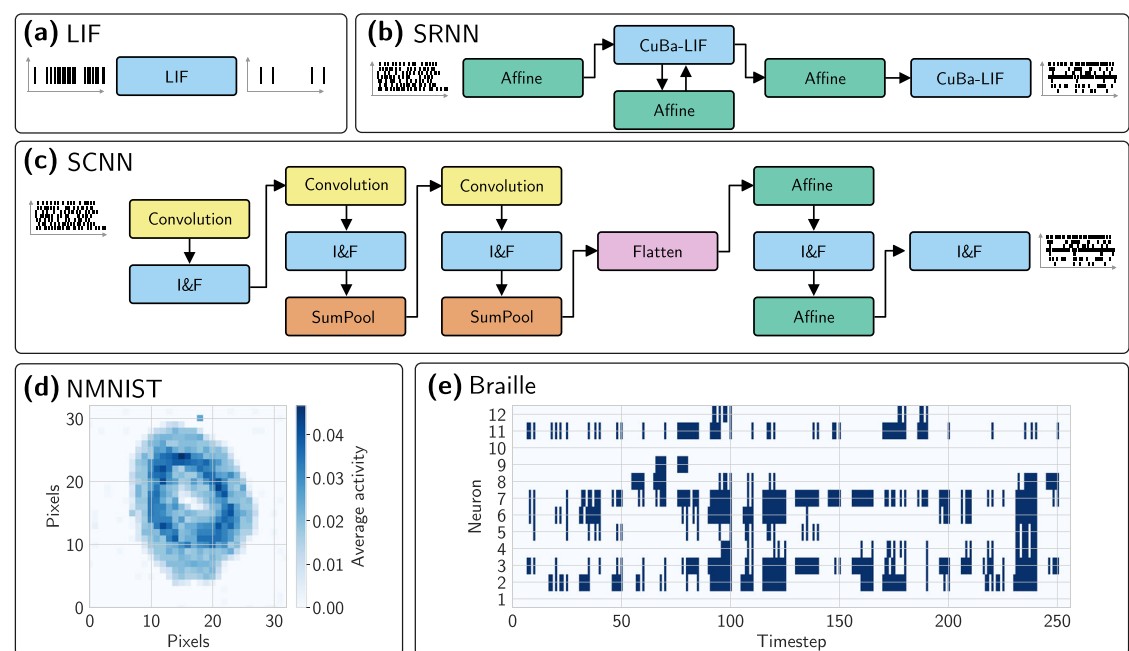

**Fig. 3 | Computational graphs and sample data used in the experiments. a** A single leaky integrate-and-fire neuron model (LIF), (**b**) a recurrent Braille classification model using current-based leaky integrate-and-fire (CuBa-LIF) in a spiking recurrent neural network (SRNN), and (**c**) a spiking convolutional neural network (SCNN). **d**, **e** shows sample data for the N-MNIST and Braille datasets, respectively. The N-MNIST activity is averaged across 300 timesteps.

three distinct tasks that are common within neuromorphic computing, through which we assess and compare performance across all compatible platforms: a single leaky integrate-and-fire (LIF) neuron, a spiking convolutional neural network (SCNN), and a spiking recurrent neural network (SRNN) (see Fig. 3).

The single LIF neuron task is a simple task that allows us to assess the visual similarity of the computations on different simulators and hardware platforms. The SCNN and SRNN represent widespread types of neural networks. Whereas CNN-based networks are very common for visual data, RNN show their strengths in

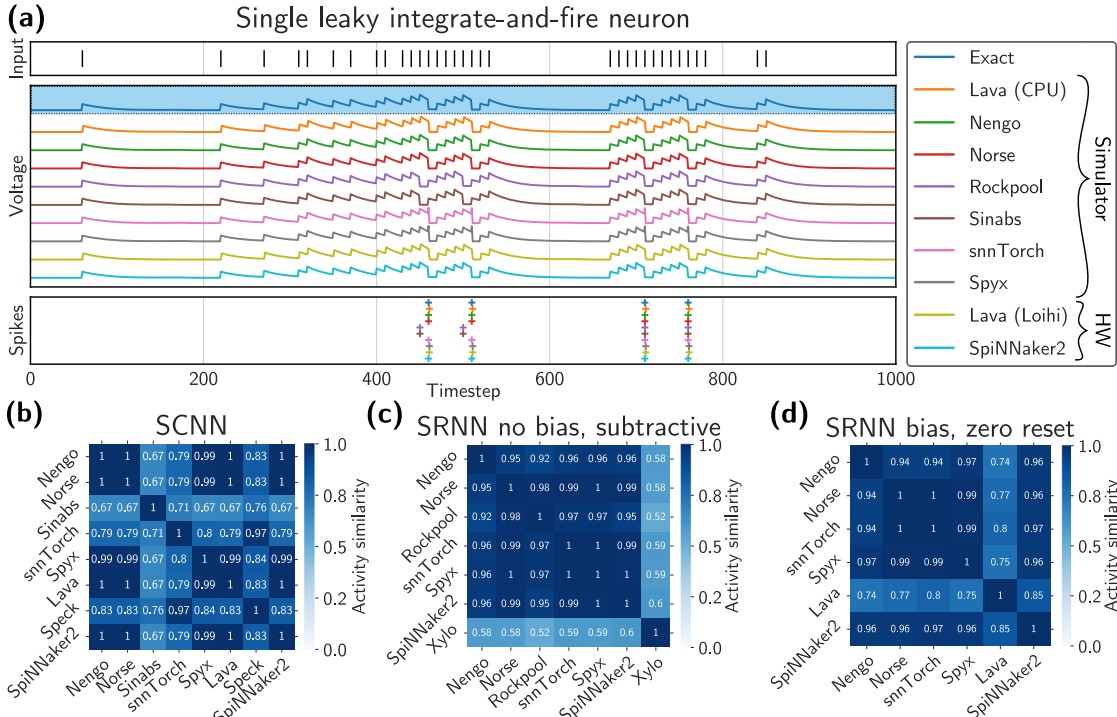

**Fig. 4 | Experimental results. a** Single leaky integrate-and fire neuron ordered from top to bottom on different platforms: input spikes, voltage traces, and output activations. The timestep of spikes is well-aligned and only differs systematically for Rockpool and Sinabs due to discretization differences. Membrane potentials are normalized to lie between the resting potential $v_{rest} = 0$ and the firing threshold $\vartheta$ to ignore platform-specific details regarding the numerical representations. **b** Spiking convolutional neural network: a platform-by-platform comparison of the spiking activity from the first spiking layer using cosine similarity (1 equals perfect overlap). Sinabs, snnTorch, and Speck deviate from most other implementations due to their discretization choices. **c, d** Spiking recurrent neural network: similarity measure between the spiking activity of the first CuBa-LIF layer for an SRNN with biases and reset to zero and an SRNN without biases and subtractive reset. See the main text for details on the similarity metrics.

tasks involving temporal relations with a limited number of input dimensions.

A central difficulty in transferring a computation from one neuromorphic platform to another is that the neuron models and discretization schemes do not always match up exactly. There will be variation between differently implemented neuron models, even for the seemingly simple LIF neuron. We refer to Section Discussion of Mismatches for a more detailed discussion of the experimental mismatches and the Supplementary Material, Section A, for the subtle differences in neuron models supported by NIR.

**Leaky integrate-and-fire dynamics.** As a first experiment, we demonstrate the behavior of a single leaky integrate-and-fire neuron under an identical input spike train (Fig. 4a) for every available platform. The LIF neuron was first instantiated in Norse, then exported into a NIR graph, and subsequently imported and executed by all different platforms. Figure 4a visualizes both the recorded voltage traces and output spikes from the platforms.

To assess the viability of NIR as an intermediate representation for neuromorphic computing, we ensure that the dynamics are consistent across all platforms regarding the most common ways of encoding information in spiking neural networks—namely, rate encoding and spike time encoding[60,64]. The equivalence of output firing rates across different platforms can easily be seen, as the number of output spikes is identical across all platforms and since the spiking timings mostly coincide. Rockpool and Sinabs notably differ because of their integration scheme. They integrate the incoming signal ($v^-$, equation (1)) when checking for a threshold crossing before they update the leak term ($v^+$, equation (2)), whereas most other simulators immediately integrate the leak (equation (3)). This effectively brings the neuron

closer to the firing threshold earlier on, as observed in Fig. 4a.

$$v^-(t+1) = \frac{dt}{\tau}\left(v(t) + v_{leak} + Ri(t+1)\right) \quad (1)$$

$$v^+(t+1) = v(t) - v^-(t+1) \quad (2)$$

$$v(t+1) = \frac{dt}{\tau}\left(v(t) + v_{leak} + Ri(t+1)\right) \quad (3)$$

We further observe that the spike times are systematically delayed by one timestep for Lava and Loihi 2. This is because Lava calculates whether a neuron spikes at timestep $t$ based on its membrane potential at timestep $t-1$, i.e., before the addition of the input to the membrane, rather than after the input was added to the membrane potential. This systematic difference will only have an effect if absolute spike timing is used, instead of relative spike timing, that is, if information is encoded in the timing between spikes.

Xylo and Speck are omitted since they do not support the leaky integrate-and-fire primitive.

**Convolutional neural network.** For the second experiment, we studied a 9-layer spiking convolutional neural network (SCNN) using 2-dimensional convolutions, sum pooling, and integrate-and-fire neurons, shown in Fig. 3c. The SCNN was trained on the Neuromorphic MNIST (N-MNIST) dataset to recognize digits from 0 to 9[65]. Details on the training and exact network architecture are shown in Section Spiking convolutional neural network. In our experiment, we compare

**Table 1 | Experimental accuracies**

| | Simulator | | | | | | Both | Hardware | | |
|---|---|---|---|---|---|---|---|---|---|---|
| | Nengo | Norse | Rockpool | Sinabs | snnTorch | Spyx | Lava* | Speck | SpiNNaker2 | Xylo |
| **SCNN** | 98.1% | 98.1% | N/A | <u>98.5%</u> | 97.9% | 97.1% | 98.2% | 95.4% | 98.2% | N/A |
| **SRNN** (Subtr.) | 78.6% | 93.57% | 71.4% | N/A | <u>92.1%</u> | 92.12% | N/A | N/A | 93.57% | 85.71% |
| **SRNN** (Zero) | 55.7% | 94.29% | N/R | N/A | <u>95.0%</u> | 84.29% | 48.6% | N/A | 85.00% | N/A |

Performance for the spiking convolutional neural network (SCNN) and the spiking recurrent neural network (SRNN) was measured against the unseen test dataset. The platform on which the spiking neural network was originally trained is underlined. * The Lava column stands for both the Loihi 2 chip and the Lava simulator since they are bitwise accurate.

not only the end-task accuracy but also the activity of the first hidden layer.

The accuracies of the SCNNs across the different platforms on the withheld test dataset are similar (see Table 1), with a mean test accuracy of 97.7% and a standard deviation of 0.9% across all platforms. These results demonstrate that NIR can effectively act as an intermediate representation of a trained, rate-based spiking convolutional neural network, as the end-task performance is preserved across all tested platforms.

We further analyze the activities of the SCNN's hidden layers across all different platforms. For a visually interpretable comparison, we use the cosine similarity of time-averaged activities, i.e., spike rates, for the first channel of the first hidden layer of I&F neurons in Fig. 4b: $S_c(\mathbf{r}_1, \mathbf{r}_2) = (\mathbf{r}_1 \cdot \mathbf{r}_2)/(\|\mathbf{r}_1\|_2 \|\mathbf{r}_2\|_2)$ where $\mathbf{r}_i$ is the flattened vector of spike rates from platform $i$. The similarity measure tells us that most platforms have almost identical dynamics, while particularly Rockpool, Sinabs, snnTorch, and Speck deviate. Since the accuracies are relatively close, we observe that the rate encoding and feed-forward architecture of the investigated SCNN are relatively robust to these mismatches.

Rockpool and Xylo are omitted from this comparison because neither support convolutions, and Xylo does not implement the IF neuron model used in the SCNN.

**Recurrent neural network.** For the third and final experiment, we studied a spiking recurrent neural network (SRNN) with one hidden layer, with populations of current-based LIF (CuBa-LIF) neurons. The network was trained in snnTorch with backpropagation-through-time (BPTT) and surrogate gradients to perform a Braille letter recognition task[66]. Further details are reported in Section Spiking's recurrent neural network. After training, the network was exported to NIR, and evaluated on all other platforms. The accuracies and similarity measures for their respective activations are shown in Fig. 4c, d. We trained two networks with different choices for the discretization of the reset mechanisms for post-spike membrane reset (shown in equations (1) and (2) in the Supplementary Material). Similar to the SCNN experiment, we measured not only the end-task accuracy but also the activity of the hidden recurrent layer.

In this experiment, the performance across different platforms was notably less consistent than in the prior two. Contrarily, the SRNN presented here displayed significant disparities in end-task performance when implemented on different platforms. This divergence can be attributed to the recurrent connections within the network, which tend to accentuate subtle discrepancies in the dynamics. The SRNN's training in snnTorch suggests that its dynamics are intricately calibrated to the specifics of this simulator. Therefore, deploying the network on a platform with dissimilarities from snnTorch's nuances can introduce pronounced differences in the system's dynamics, thereby affecting the end-task outcome.

In sum, NIR emerges not only as a representation but also as a measurement tool—it facilitates a deeper exploration of the discrepancies between platforms, revealing the influence of diverse parameterization, levels of precision, and models on the resultant performance mismatches. NIR thus offers a valuable avenue for further research by quantifying the robustness of neural networks, specifically

gauging their resilience to simulator and device mismatch, and thereby stands as a pivotal tool for the advancement of neuromorphic computing.

**Discussion of mismatches.** Instead of providing perfect functional reproducibility, NIR establishes an idealized model that serves as a common ground for comparative analysis while highlighting inter-platform discrepancies. We attribute these discrepancies to three main causes:

**Neuron model implementation.** As shown in Section A in the Supplementary Material, the simulators and chips do not use the same neuron model definitions. Whereas some definitions can be made equal by matching parameters, differences in spike and reset timing or different discretization choices lead to changes in neuron dynamics.

**Quantization.** The neuromorphic chips we tested use quantization to reduce on-chip memory requirements and power required for computation. Quantization will change the activity due to inherent rounding errors, which can cause performance degradation[67]. We only used post-training quantization to reveal discrepancies between the frameworks. Although quantization-aware training (QAT) or fine-tuning may be able to recover lost performance, we do not provide a detailed study of the extent to which QAT reduces mismatches here.

**Determinism.** Whereas simulation frameworks are usually deterministic in their computation, asynchronous hardware, such as Speck and SpiNNaker2, are not fully deterministic. This variability can lead to further activity discrepancies across platforms. Most event-based processing systems do not guarantee the determinism of the individual events.

## NIR axioms

The nodes in NIR are described by hybrid continuous-time systems, capturing the realistic evolution of some physical quantity as well as jumps and discrete resets. Instead of executing these dynamics, the NIR graph merely declares them, ensuring that any computational backend is equipped with the complete information required to interpret and evaluate the represented computation. It is essential to clarify that NIR's objective is not to rectify discrepancies arising from mismatches between platforms. Rather, NIR offers a common framework on which all hardware platforms can address shared challenges and brings into sharp focus the inconsistencies that arise from a heterogeneous neuromorphic computing landscape where neuron models and discretization schemes are not standardized.

**Axiom 1.** (Neuromorphic computational primitives) NIR provides a set of standardized computational primitives $\mathcal{C}$. Each computational primitive $c \in \mathcal{C}$ defines a parameterized transformation $T_\theta$ of a continuous-time input signal $u(t)$ into a continuous-time output signal $y(t)$, where $\theta$ is the set of parameters for a given compute primitive $c$.

(1.1) Every computational primitive $c$ is modeled as a hybrid system that describes how the computational dynamics evolve over

**Table 2 | Computational primitives in NIR**

| Primitive | Parameters | Parameter types | Computation |
|---|---|---|---|
| Input | Input shape | $\mathbb{N}^{N_0 \cdots N_n}$ | – |
| Output | Output shape | $\mathbb{N}^{N_0 \cdots N_0}$ | – |
| Affine | $W, b$ | $\mathbb{R}^{N_0 \cdots N_n} \times \mathbb{R}^{N_1 \cdots N_n}$ | $W\,i(\cdot) + b$ |
| Convolution | $W$, Stride, Padding, Dilation, Groups, Bias | $\mathbb{R}^{C_{out} \times C_{in} \times N_0 \cdots N_n} \times \mathbb{N}^{N_0 \cdots N_0} \times \mathbb{N}^{N_0 \cdots N_0} \times \mathbb{N}^{N_0 \cdots N_0} \times \mathbb{N} \times \mathbb{R}^{N_0 \cdots N_0}$ | $f \star g$ |
| Delay | $\tau$ | $\mathbb{R}^N$ | $i(t - \tau)$ |
| Flatten | Input shape, Start dim., End dim. | $\mathbb{N}^{N_0 \cdots N_n} \times \mathbb{N} \times \mathbb{N}$ | - |
| Integrator | $R$ | $\mathbb{R}^{N_0 \cdots N_n}$ | $\dot{v} = R\,i(\cdot)$ |
| Leaky integrator (LI) | $\tau, R, v_{leak}$ | $\mathbb{R}^{N_0 \cdots N_n} \times \mathbb{R}^{N_0 \cdots N_n} \times \mathbb{R}^{N_0 \cdots N_n}$ | $\tau\dot{v} = (v_{leak} - v) + R\,i(\cdot)$ |
| Linear | $W$ | $\mathbb{R}^{N_0 \cdots N_n}$ | $W\,i(\cdot)$ |
| Scale | $s$ | $\mathbb{R}^{N_0 \cdots N_n}$ | $s\,i(\cdot)$ |
| Spike | $\theta_{thr}$ | $\mathbb{R}^{N_0 \cdots N_n}$ | $\delta(i(\cdot) - \theta_{thr})$ |
| **Higher-order primitive** | | **Composition** | |
| Integrate-and-fire (I&F) | | | |
| Leaky integrate-and-fire (LIF) | | | |
| Current-based leaky integrate-and-fire (CuBa-LIF) | LIF ∘ Linear ∘ LI | | |

$$\text{(Leaky) Integrator} \xrightarrow{\hspace{2em}} \text{Spike}, \quad \overset{\longleftarrow \text{Linear} \longleftarrow}{\phantom{x}}$$

Top: The computational primitives, their parameters, and computational models. Bottom: Higher-order primitives as compositions. Parameters are typed for an arbitrary number of dimensions, $n$. $i(\cdot)$ denotes input at a given point in time, where $t$ denotes a specific time, $C_{out} \times C_{in}$ denotes convolutional output and input channels, and $\delta$ denotes the Dirac delta function.

time. This includes differential equations ($\dot{x} = f_\theta(x, u)$) as well as discretized transition functions for sudden activations, where $x(t) \in \mathbb{R}^{N_x}$, $u(t) \in \mathbb{R}^{N_u}$ are multidimensional signals. For brevity, we represent the set of parameters for a particular compute primitive with $\theta$, although this set of parameters varies for every primitive (see Table 2). For example, a leaky integrator may be modeled with the ODE $\dot{x} = -x + u$, and a linear connection layer may be modeled with the transfer function $y = Wx$.

(1.2) Every primitive offers a set of input ports $p_{in} \in \mathcal{P}$, and a set of output ports $p_{out} \in \mathcal{P}$. A single port consists of a name and the expected shape of signal, $\mathcal{P} = (\mathcal{S} \times \mathbb{N}^n)$ where $\mathcal{S}$ is the set of strings and $n$ is the number of independent signal channels. By exposing one or more ports, a single computational primitive can receive multiple input signals $u(t) = (u_1(t), \ldots, u_{n_u}(t))$ and send multiple output signals $y(t) = (y_1(t), \ldots, y_{n_y}(t))$, effectively separating semantically different signals. For example, a neuron may receive one input signal that models a synaptic current input and another input signal that serves as a neuromodulatory signal. Ports are uniquely identifiable via a name, which, in the case of input ports, is similar to the argument name of a function.

(1.3) Computational primitives may be composed into higher-order primitives, as shown at the bottom of Table 2 and illustrated in Fig. 2.

**Axiom 2.** (Graph-based computation) Every computation is represented as a NIR graph $\mathcal{G} = (\mathcal{V} \times \mathcal{E})$ where $\mathcal{V}$ is the set of computational nodes (e.g., a leaky integrate-and-fire neuron, or a linear connection layer) and $\mathcal{E}$ is the set of directed connections between the nodes in the graph.

(2.1) A computational node $v \in \mathcal{V}$ is composed of a primitive $c \in \mathcal{C}$ and its parameters $\theta$. This yields a complete description of an input-output transformation, and thus fully describes the computation implemented by the computational node $v$. For example, the computational node representing an affine linear map consists of the primitive $c_{Affine}$ along with the weight matrix $W$ and the bias $b$ as the parameters $\theta = (W, b)$, such that $v_{Affine} = (c_{Affine}, \theta)$.

(2.2) An edge is a directional connection from an outgoing port to an incoming port $\mathcal{E} = (\mathcal{V} \times \mathcal{P}_{in}) \times (\mathcal{V} \times \mathcal{P}_{out})$, where $\mathcal{P}_{in}$ and $\mathcal{P}_{out}$ are the ports of the input and output nodes, respectively. There are no restrictions on the number of connections any node or port can have. As an example, the edge $e = ((v_{Affine}, p_{out}), (v_{Affine}, p_{in}))$ connects the output of the affine node to the input of that same node.

(2.2.1) Edges do not perform any computation and are effectively the identity map $I: x \mapsto x$.

(2.2.2) One input port can receive multiple incoming edges, which are then, by convention, summed together element-wise. Similarly, one output port can send its information via different edges to various input ports. The information is then passed by values, not by reference, such that information can only flow along the direction of an edge.

The above axioms closely follow conventional descriptions of directed signal-flow graphs. Figure 2 further illustrates the workings of NIR graphs, and Fig. 3 shows three example NIR graphs that are used in the experiments from Section Experiments.

## Computational primitives

NIR defines 11 computational primitives and 3 higher-order primitives, listed in Table 2, where each primitive describes the evolution of a hybrid system over time (see Section NIR axioms). We define common neuromorphic components like the linear map, leaky integrator, and spike threshold function, but we further included mathematical primitives such as the affine map and convolution. The input and output nodes serve to disambiguate the entries and exits of a graph.

The 11 primitives in NIR are "fundamental" in the sense that the backends implementing the primitives are required to approximate the computation of the idealized description as closely as possible within the limitations of the platform. Any given platform is not expected to implement the full specification. This is particularly true for functionally specialized hardware, where hardware restrictions render certain functional primitives impossible. We

detail such restrictions in Section Hardware platforms and how NIR can deal with hardware constraints in Modeling hardware constraints with NIR.

NIR does not restrict the implementation apart from stating the idealized descriptions, and it is expected that the exact computation of the primitives will vary across platforms. This is particularly important for digital systems and their choice of integration and discretization methods. Given the heterogeneity of the present hardware landscape, this amount of freedom becomes a practical choice as well; imposing any restriction on the integration scheme for the hybrid ODEs will invariably lead to incompatibilities and numerical deviations across platforms.

The primitives listed in Table 2 can be composed to provide more complex computational elements. Three such examples are already present in the specifications as convenient abstractions:

1.  The integrate-and-fire (IF) neuron is defined as an integrator composed of a spike function and a feedback mechanism that acts as a membrane reset.
2.  The leaky integrate-and-fire (LIF) neuron is defined as a leaky integrator composed of a spike function and a feedback mechanism that acts as a membrane reset.
3.  The current-based leaky integrate-and-fire neuron (CuBa-LIF) is defined as a leaky integrator, linearity, and leaky integrate-and-fire neuron.

**Relation between NIR and existing intermediate representations**
The field of neuromorphic computing is divided into separate paths for digital and analog hardware developments, which often progress in isolation. By providing a unified framework that accommodates both of the paths, NIR transcends this divide, capturing the underlying essence of neuromorphic computing that is rooted in the use of time as a computational element and the exploitation of continuous-time dynamics where "time represents itself"[68]. To address these aspects, NIR draws inspiration from and shares commonalities with, previous efforts. Below, we sketch the unique characteristics of NIR in relation to other intermediate representations related to neuromorphic computing. A tabular overview is available in the Supplementary Material, Section B.

**Modular approach to primitives.** NIR stands out by not prescribing a single representation for computations but instead offers a modular approach where primitives can be composed arbitrarily. The concept of composing primitives has proven successful in other frameworks like Fugu and Lava for the neuromorphic domain, and Compiler infrastructure projects like LLVM[56] and MLIR[7] for the conventional domain. Both LLVM and MLIR are designed to optimize discrete machine-level instructions, rather than continuous-time systems equations. Modularity and flexibility allow NIR graphs to be tailored to diverse hardware platforms, ensuring that they can be adapted to the constraints of each platform (see Section Modeling hardware constraints with NIR). Moreover, NIR expands the application of neuromorphic computing beyond neuroscience to include deep learning operations, enabling the modeling of both artificial neural networks (ANNs) and spiking neural networks (SNNs), as well as supporting hybrid ANN-SNN computations.

**Agnostic to hardware/software platforms.** NIR adopts a purely declarative style, making it agnostic to the specifics of any hardware or software platform. Any given backend can choose to interpret and implement NIR graphs, as long as the backend approximates NIR's underlying continuous-time dynamics. This contrasts efforts like PyNN, NeuroML, and Fugu, which make assumptions about the runtime environments. NIR is designed to integrate with existing standards and compilers. This interoperability aligns NIR with model-centric APIs in deep learning and frameworks like MMdnn[11] or Ivy[10],

facilitating the transfer of models and computations across different libraries and platforms.

**Serializable exchange format.** Currently, NIR functions primarily as a representation and exchange format, focusing on the portability of core model dynamics rather than optimizing hardware efficiency and model performance. This approach positions NIR similarly to ONNX[6] or MMdnn[11], contrasting with frameworks like MLIR[7] and TVM[9] that provide cross-compilation and multi-layer abstractions. As NIR's base representations stabilize, there is potential to introduce additional layers of abstraction, enhancing its capability to serve as a foundational building block for future computational frameworks.

**Applicability to multiple time scales.** NIR and NeuroML model continuous-time systems, which translate trivially to continuous-time simulators, as shown in Fig. 1a. By supporting different time constants in the same computational graph, the model can detail dynamics at multiple time scales simultaneously. Behaviors at multiple levels are challenging to capture with discrete simulators that advance via fixed-time steps, particularly if they are complex or chaotic.

## Discussion

We presented an intermediate representation for digital neuromorphic systems, the Neuromorphic Intermediate Representation (NIR), as a platform-independent set of continuous-time computational primitives. NIR can be viewed as a "neuromorphic instruction set" that can be composed arbitrarily. It is an ideal serializable representation for multiple neuromorphic platforms, serving as a storage format for software platforms and a source language for these hardware systems. We documented the support of NIR across 11 different neuromorphic platforms on which we demonstrated three specific computational graphs: a leaky integrate-and-fire neuron, a spiking convolutional neural network using integrate-and-fire neurons, and a spiking recurrent neural network using current-based leaky integrate-and-fire neurons. We found that NIR faithfully represents the idealized underlying computation and that the execution aligns well across the supported platforms for feed-forward dynamics, such as a single LIF neuron or a spiking convolutional network. For more complex and time-sensitive models, such as the recurrent network, we found that platform-specific constraints and discretization choices produced somewhat diverging activation patterns and accuracies. Although the graphs are compatible, the continuous-time nature of the primitives requires platform-dependent optimization, such as quantization-aware training, on digital systems to achieve the highest accuracies. The current version of NIR excludes other important mechanisms such as adaptive threshold mechanisms[69], gating[70,71], resonate-and-fire[62,72], and multicompartmental[73,74] neuron models. That said, the open design of NIR allows for easy integration of new primitives, ensuring that NIR can adapt to new applications based on neuronal computing without changing other system-level aspects.

In the context of neuromorphic computing, the notion of mismatch has traditionally been associated with the imperfections and variability inherent in analog hardware systems. However, our findings highlight a crucial and often overlooked aspect of mismatch in purely digital neuromorphic systems, particularly when working with recurrent neural networks. This underscores the need for strategies that address mismatch across both analog and digital neuromorphic systems to ensure consistent computational outcomes and enhance system reliability across various applications.

One key aspect that we do not discuss in this work is the connection to physical systems and future hardware platforms, including analog. NIR can represent models of arbitrary scale, limited only by the underlying hardware of (1) the system that converts a model to and from the NIR format, and (2) the neuromorphic hardware that runs such a model. Both the SpiNNaker2 and Loihi platforms scale to multi-

chip systems, which is necessary to address larger problems such as energy-efficient large models[75]. For example, Hala Point combines 1.15 billion neurons distributed across over 1k Loihi 2 processors. In addition, NIR can be composed into highly complex systems of equations, such as higher-order systems, which lends itself well to the study of diffusion, fluid dynamics, and optimization problems for a given architecture-hardware-software system. NIR primitives can be realized directly as physical circuit components and can, therefore, generalize to a richer set of hardware systems than the ones we have discussed so far. Mapping such models onto larger and richer hardware platforms would be exciting to explore in future work.

It is crucial that the computations represented by NIR graphs remain the same, despite evolving primitives and future developments. An important step towards this is the continued testing of platform integrations along with a well-documented versioning approach for the primitives akin to the Operator Set (opset) number in ONNX[6]. In addition, and specifically for digital hardware, it would be interesting to explore the integration of lower-level hardware details for multi-level models that further optimize efficiency, similar to MLIR dialects[7].

The current computational experiments were chosen to demonstrate the capabilities of NIR as a proof-of-concept of the representation's modular functionality. Hence, tasks such as N-MNIST with SCNN and Braille dataset with SRNN are not challenging for current ML algorithms, and we prioritized these illustrations on state-of-the-art neural network architectures. Yet, the primitives used in the models are highly relevant for neuromorphic computing[76], since recent work on CNNs and ResNets is successfully competing with ANNs for vision[77–79], and for time-series classification[80]. These SNNs use far less computing than their ANN counterparts, which can significantly reduce energy consumption when implemented on dedicated hardware[81]. Large-scale neuromorphic systems like Loihi or SpiNNaker2 and others have shown orders of magnitude energy and latency improvements compared to off-the-shelf CPUs and GPUs, see refs. [82,83] for reviews. As the deep learning community has moved to transformers as a primary component, also efficient, brain-inspired architectures such as SpikeGPT[84] have been developed. There may be specific applications that run on these current architectures that NIR may add complex operations, which may reduce energy efficiencies. For these systems, we do not recommend using NIR as a means to increase efficiencies. However, for many applications, NIR can become a useful tool for cross-platform interoperability by providing flexibility in choosing the most efficient hardware platform for any given neural architecture. In addition, depending on the application, NIR may reduce the number of training operations with a potential increase in energy efficiency.

Our main goal has been to address the lack of shared representations for neuromorphic computing. The deployment of software-generated models on neuromorphic hardware was previously only possible through hardware-specific software libraries. NIR effectively decouples the development of neural models from platform-specific tools, such that software and hardware can evolve independently, as shown in Fig. 2, and such that models developed in one framework can be readily deployed and reproduced across neuromorphic simulators and hardware within device-specific boundary conditions. In turn, NIR allows users to directly leverage tools and resources from the active

and growing neuromorphic open-source community, such as quantization mechanisms or algorithms for fast training. Furthermore, the interoperability of spiking neuron models simplifies benchmarking and comparison across platforms. Any framework that chooses to integrate with an intermediate representation, such as NIR, benefits from these tools as well as a much larger user base compared to standalone technology stacks. The present set of primitives is limited, and many platforms are not included in our analysis. However, similar to how the initial design of digital instruction sets propelled the advent of modern-day computing, we believe this to be a crucial first step towards further development of shared tooling infrastructure for neuromorphic computing. We have argued that abstracting away hardware-specific details will lower the barrier to entry for new neuromorphic hardware systems, and it is our hope that NIR can accelerate the academic development required to propel neuromorphic computing to perform on par with present-day deep learning while solving practical problems in the near future.

## Methods
### Computational graphs
Computational graphs are interconnected computational nodes through which signals flow and have been an important tool for computational studies in analog and digital systems since their inception in the 1940s[53]. In its most abstract form, computational graphs ($\mathcal{G} := (\mathcal{V}, \mathcal{E})$) consist of a set of nodes ($\mathcal{V}$) and edges ($\mathcal{E} := (\mathcal{V} \times \mathcal{V})$) connecting two nodes. An example of a computational graph could be discrete components on a circuit board, where wires connect components together. Any state needed in the individual components, such as voltage, memory, etc., would be dealt with internally. Another example could be layers in a machine learning model, where the order in which the layers are applied determines the flow of the input signal through the model. Most machine learning models do not require a state contrary to the circuit example, with the exception of some algorithms such as batch normalization or recurrent neural networks.

In the 1950s Mealy introduced a notation that explicitly captures state, as it would have to be stored in, say, digital circuits[55]. Figure 5 visualizes the Mealy machine on the right side, where some input signal, $X$, is transformed according to some ruleset in $T$ that recursively operates on the state $S$. Both $S$ and $X$ are then transformed according to another ruleset $O$ which computes the final output $Y$. Mealy machines formally separate memory (state) from computation (nodes). The same distinction cannot be made in analog systems, where the system itself represents the state computation. While signal flow graphs are technically more ambiguous regarding the representation of the state, they compactly represent computational graphs for both digital and analog systems.

In the following, we restrict ourselves to the domain of first-order systems, described by ordinary differential equations (ODEs) of the form

$$\dot{x} = f(x; \theta), \quad x \in \mathbb{R}^N, \theta \in \mathbb{R}^M \tag{4}$$

where $x(t)$ is a real-valued function of time $t$, $\dot{x}$ is the derivative of x with respect to time, and $f(x)$ is a continuous real-valued function of the $N$-dimensional input $x$. $\theta$ describes a fixed set of $M$-dimensional

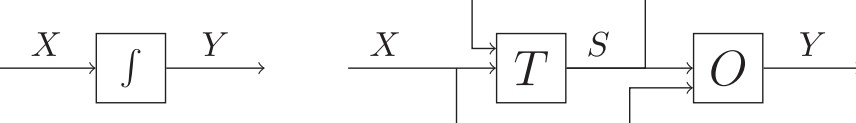

**Fig. 5 |** Left: a signal flow graph where a signal $X$ is integrated by some arbitrary process to produce the output signal, $Y$. Right: a Mealy machine where a transition node $T$ updates a recurrent state $S$ that is forwarded to an output node $O$ which, in turn, determines the output $Y$.

parameters. $f$ may be smooth, but it may also be subject to sudden forces, such as a bouncing ball hitting a hard surface. For that reason, we use hybrid systems, defined as the combination of continuously smooth functions as well as functions with discrete transitions. We explicitly model jumps by allowing conditional differential equations of the form

$$z = \begin{cases} f & \text{Condition} \\ g & \text{Else} \end{cases} \qquad (5)$$

where $f$ and $g$ are real-valued functions in time.

Digital systems require discrete instructions and cannot directly solve the equations above. Numerical methods for optimally discretizing continuous ODEs are a mature and active research area with a long history. Despite the discrepancy between the continuous-time ODE formulation and digital computers, modern fixed-point neural solvers allow for sound accuracy with relatively large timesteps and low precision[85]. Therefore, we argue that the continuous-time ODE formulation is not a hindrance for digital systems. In fact, it allows NIR to express computations at a higher level of informational abstraction that is independent of how the system is discretized. This may result in inaccuracies across systems that we addressed in Section Computational primitives.

### Modeling hardware constraints with NIR

Not all NIR graphs can be executed by every hardware platform, due to hardware constraints. We can express these as constraints on the computational graphs that the hardware supports. For example, the Xylo Audio 2 chip limits the model to a size of up to 1000 LIF neurons, each with a maximum fan-out of 64. This means that a given NIR graph $\mathcal{G}$ is supported by Xylo only if it contains $\leq 1000$ LIF neurons, each with $\leq 64$ incoming connections.

In this simple example, the constraints are easy to verify, see Box 1. In generality, however, this is a difficult problem that is related to the NP-complete subgraph isomorphism problem. To illustrate the difficulty of this problem, consider a hardware platform that supports only linear connections, i.e., $y = Wx$, and no affine connections, i.e., $y = Wx + b$. We have a computational graph $\mathcal{G}$ in NIR that contains an affine connection node. A naïve constraint checking algorithm, like the one shown in Box 1, would decide that the NIR graph $\mathcal{G}$ is incompatible with the hardware. However, if the bias $b$ of the affine layer equals zero, we could map the graph $\mathcal{G}$ to an "isomorphic" graph $\mathcal{G}'$, in which the affine connections with zero biases are replaced by linear connections.

Because NIR allows for the composition of existing primitives, mapping an NIR graph to another platform involves pattern-matching subgraphs against platform-compatible primitives. For example, snnTorch represents a recurrently connected LIF population as a separate building block called `RLeaky`. Thus, to convert an RNN from NIR to snnTorch, we must detect subgraphs that are isomorphic to `LIF ↔ Linear`. Moreover, higher-order primitives are obviously isomorphic to the composition of lower-level primitives that define them (see Table 2).

If we allow for approximations in mapping the graph to the hardware, this problem gets further complicated as we could, for example, replace a LIF neuron with a CuBa-LIF neuron with a synaptic time constant approaching $\tau_{syn} \to 0$.

### Neuromorphic simulators and hardware platforms

As illustrated above, NIR is compatible with many simulators and hardware platforms that implement the de facto dynamics of the underlying hybrid ODEs. Specifically, we have made a series of choices around the numerical integration of the systems of equations that we detail alphabetically below.

### Hardware platforms

**Intel Loihi 2**. The Loihi 2 chip by Intel consists of 6 embedded microprocessor cores (Lakemont x86) and 128 fully asynchronous neuron cores (NCs) connected by a network-on-chip, as explained in[86]. The NCs are optimized for neuromorphic workloads by implementing a group of spiking neurons and including all synapses connected to such neurons. All the communication between NCs is in the form of spike messages. Microprocessor cores are optimized for spike-based communication and execute standard C code to assist with data I/O as well as network configuration, management, and monitoring. Some new functionalities added in this second version of the Loihi chip are the possibility of implementing custom neuron models using microcode instructions (assembly), the option to generate and transmit graded spikes, and support for three-factor learning rules. A single Loihi 2 chip supports up to 1 million neurons and 120 million synapses. Together with Loihi 2, Intel presented their open-source framework Lava, that allows users to write neuro-inspired applications and map them to both traditional and neuromorphic hardware. Using high-level Python APIs, users can describe their neural networks, which are then compiled to run on the requested backend. Currently, Lava supports deployment on traditional CPU and Loihi 2. Specifically for Loihi 2, Lava also gives the possibility of writing custom neuron models in assembly to be run on the NCs, and custom C code to be run in the microprocessor cores. Because of its programmability, the Loihi 2 chip supports different precision levels for the quantization of parameters and activations. For our experiments, we used 24 bits for state variables, 16 bits for activations, 12 bits for time constants, 17 bits for the thresholds, and 8 bits for the weights.

**SpiNNaker2**. SpiNNaker2 is an IC designed for the simulation of very large-scale spiking neural networks. In contrast to many other solutions, it uses 152 processing elements (PE) connected by a network on a chip. Each PE contains an ARM Cortex M4F core with 128 kB local SRAM, as well as accelerators for exponential and logarithm functions and a $16 \times 4$ MAC array for 2D convolution and matrix-matrix multiplications[87]. Multiple chips can be connected using dedicated chip2chip links and an on-chip packet router optimized for small packets of spikes. In addition, each chip can be connected to LPDDR4 DRAM in order to extend the amount of memory available per chip. The py-spinnaker2 software framework[88] uses 8-bit signed synapse weights and 32-bit floating-point numbers for neuron parameters and state variables, respectively. Currently, IF, LIF and CuBa-LIF neuron models are implemented and supported, both reset by subtraction or reset to zero. Since SpiNNaker2 is software-based, further models can be added. It is also not limited to SNN execution but can be used for any computational task, including real-time control or deep neural networks.

## BOX 1

# Pseudocode for naive verification of the compatibility of a NIR graph with the Xylo chip

```
1  def is_compatible_with_xylo(g: nir.NIRGraph):
2      lif_nodes = filter(is_lif, get_leaf_nodes(g))
3      if len(lif_nodes) > 1000:
4          return False
5      for lif_node in lif_nodes:
6          pre_nodes = get_pre_nodes(lif_node, g)
7          if len(pre_nodes) > 64:
8              return False
9      #... (other constraints)
10     return True
```

**SynSense Speck**. Speck is an integrated sensor-processor IC that fuses event-based vision sensing with event-driven spiking CNN processing. The ultra-low-power IC operates fully asynchronously and takes full advantage of the asynchronous nature of events produced by the DVS. These events are processed by Integrate and Fire neurons that are interconnected efficiently using a convolutional engine within each core. The chip version used in this work comprises of 9 dedicated SCNN cores, each consisting of convolutional connections, IF neurons, and pooling. The chip supports 8-bit weight resolution and 16-bit membrane resolution. Each core supports convolutions of up to 1024 input and output channels with strides of 1, 2, 4, or 8. The device is accessible via a high-level Python library Sinabs or a low-level library samna. Further details are available at sinabs.ai and synsense.ai.

**SynSense Xylo**. Xylo is a series of ultra-low-power devices for sensory inference, featuring a digital SNN core adaptable to various sensory inputs like audio and bio-signals. Its SNN core uses an integer-logic CuBa-LIF neuron model with customizable parameters for each synapse and neuron, supporting a wide range of network architectures. The Xylo Audio 2 model (SYNS61201) specifically includes 8-bit synaptic weights, 16-bit synaptic and membrane states, two synaptic states per neuron, 16 input channels, 1000 hidden neurons, 8 output neurons with 8 output channels, a maximum fan-in of 63, and a total of 64,000 synaptic weights. For more detailed technical information, see https://rockpool.ai/devices/xylo-overview.html. The Rockpool toolchain contains quantization methods designed for Xylo, as well as bit-accurate simulations of Xylo devices.

**Simulation frameworks.** Most simulation frameworks mentioned here are based on the machine learning accelerator PyTorch[89]. PyTorch effectively operates as a compiler that translates Python code to various digital computing architectures, including CPUs and GPUs. This does not help the simulation frameworks in addressing the discretization problems mentioned above, although PyTorch-related features such as quantization and varying floating-point precision to approximate hardware constraints can be useful.

**Lava** is an open-source software stack developed by Intel and used for programming their Loihi 2 chip[43]. Lava has a modular structure, supporting versatile processes from neurons to external device interfaces. These processes communicate via event-based messaging and are adaptable to various platforms, including CPU, GPU and the Loihi 2 chip. Within Lava, there are several other sub-libraries. For NIR, we are using Lava itself as well as Lava-dl, which offers offline training, online training, and inference methods for various deep event-based networks.

**Nengo** is used to implement networks for deep learning, vision, motor control, visual attention, serial recall, action selection, working memory, attractor dynamics, inductive reasoning, path integration, and planning with problem-solving[46]. Nengo has been applied to cognitive modeling, deep learning, adaptive control, and accurate dynamics, and integrates with several hardware platforms, including CPU, GPU, FPGA, Loihi 2, and SpiNNaker1. While Nengo aims to be agnostic to particular neuron models by automatically locally retraining weights, here we use the neuron model of its default CPU implementation.

**Norse** is based on PyTorch and models stateless networks by explicitly passing the state of the neuron computation into each computation[57]. This approach simplifies parallelization and sharding for both the model and the data. Norse applies simple feed-forward Euler integration and implements various surrogate and adjoint methods for gradient-based optimization, as well as spike-time dependent plasticity and Tsodyks-Markram short-term plasticity for unsupervised adaptation.

**Rockpool** is a machine-learning framework for SNNs, supporting network design, training, testing, and deployment to neuromorphic hardware[58]. Rockpool provides a torch-like interface, with automatic differentiation back-ends and hardware acceleration based on PyTorch and JAX[90]. The library includes hardware-aware training, including quantization- and pruning-in-training, as well as post-training quantization. Rockpool includes a flexible and extensible deployment framework based on graph extraction, which currently includes deployment to multiple devices in the Xylo family.

**Sinabs** is a deep learning library based on PyTorch for spiking neural networks, with a focus on simplicity, fast training, and extensibility[59]. Sinabs works well for computer vision models as it supports weight transfer from conventional CNNs and enables deployment to Speck, the spiking convolutional processor. With its support for EXODUS[91], it allows for fast training of deep SNNs. It integrates seamlessly into libraries built on top of PyTorch such as Lightning.

**snnTorch** provides a thin abstraction layer on top of PyTorch for training and modeling SNNs[60]. It prioritizes flexibility by providing the option of stateless and stateful networks to the user. It integrates various learning rules, including customizable surrogate gradient descent with backpropagation through time, along with online learning rules such as real-time recurrent learning variants and spike-time dependent plasticity. snnTorch also accounts for hardware-friendly training approaches, including stateful quantization that digital accelerators may need to account for, as well as probabilistic neuron models that factor in noise models typical of analog or mixed-signal hardware.

**Spyx**[61] is a lightweight and modular package for training SNNs within the JAX ecosystem[90]. Extending Google Deepmind's Haiku library[92] for training deep neural networks, Spyx offers a host of simplified spiking neuron models and utility functions that make it easy to compose and train SNN architectures through surrogate gradient descent or neuroevolution. A notable feature is the ability for users to utilize mixed precision with minimal code modification and leverage Just-In-Time (JIT) compilation for the entire training loop to achieve maximal hardware utilization on modern deep learning accelerators such as GPUs or TPUs.

### Training setup
We proceed to describe the training setup used to obtain the architecture and parameters for the SCNN and SRNN in the second and third experiments, respectively. The training code can be found on https://github.com/neuromorphs/NIR/tree/main/paper.

**Spiking convolutional neural network.** For our SCNN task, we followed the ANN to SNN model conversion approach from[93]. First, we trained a non-spiking ReLU-based convolutional neural network on the neuromorphic MNIST dataset (N-MNIST)[65] with the following architecture: Conv (16 channels, 5 × 5 kernel, 2 × 2 stride) - ReLU - Conv (16 channels, 3 × 3 kernel, 1 × 1 stride) - ReLU - SumPool (2 × 2 kernel), Conv (8 channels, 3 × 3 kernel, 1 × 1 stride) - ReLU - SumPool (2 × 2 kernel) - Flatten - 1-layer MLP (256 hidden units, ReLU activation on hidden and output). All convolutional layers use a padding of 1 × 1, and there are no biases for the convolutional or fully connected layers. The last layer of the network contains 10 neurons, each one representing one digit.

Each sample from the N-MNIST dataset was turned into three frames by aggregating over the event count in time, thus creating 180,000 training samples and 30,000 testing samples. The ANN was trained using backpropagation to optimize the cross-entropy loss using the Adam optimizer with a learning rate of $\sigma = 0.001$. The network was trained for four epochs, after which it reached a validation loss of 0.06 and a validation accuracy of 98%. This ANN was then transferred to an equivalent spiking convolutional network with Sinabs. Hence, the neuron with the highest firing rate represents the label prediction.

**Spiking recurrent neural network.** For our SRNN task, we trained a spiking recurrent neural network (SRNN) with one hidden layer on a Braille letter recognition task[66]. Data from pressure readings acquired through an artificial fingertip and encoded using a sigma-delta modulator with $\vartheta = 1$ was used, accounting for a time binning with a bin size equal to 5 ms.

Compared to the original implementation in ref. [66], we introduced two simplifications to fit the connectivity and size constraints of the Xylo chip. First, by avoiding input copies and by collapsing the ON and OFF channels of each tactile sensor into a single spike array through an OR-like operation at every timestep, we reduced the input size to twelve. Second, we selected a subset of characters ('Space', 'A', 'E', 'I', 'O', 'U', 'Y') to make this a spatio-temporal classification problem that can be handled with only seven neurons in the output layer. For the hidden recurrent layer, 38 (zero, with bias) or 40 (subtractive, without bias) CuBa-LIF neurons are used, whereas the output layer contains seven CuBa-LIF neurons. The identification of optimal hyperparameters for the SRNN was achieved by performing an optimization procedure adapted from the one described in ref. [94].

The network was trained with backpropagation-through-time (BPTT) using surrogate gradients in snnTorch. To optimize the spiking activity of the output neurons, training was performed on the cross entropy of the spike count at the output layer. The spike function was implemented with the Heaviside function in the forward pass and approximated with the fast sigmoid function in the backward pass[95]: $S \approx \frac{U}{1+k|U|}$ with $\frac{\partial S}{\partial U} = \frac{1}{(1+k|U|)^2}$ where $k = 5$. A regularization term was added to this loss function to take into account both the $\ell^1$ norm of the average number of spikes per neuron and the $\ell^2$ norm of the total number of spikes in the recurrent layer. The coefficients for such regularization were chosen through the above-mentioned hyperparameter optimization, and the optimal values found were $\mu_{\ell^1} = 6 \times 10^{-4}$, $\mu_{\ell^2} = 4 \times 10^{-6}$ (zero, bias) and $\mu_{\ell^1} = 1 \times 10^{-3}$, $\mu_{\ell^2} = 1 \times 10^{-6}$ (subtractive, no bias). The objective was minimized using the Adam optimizer with a learning rate of 0.005 (zero, bias) or 0.001 (subtractive, no bias). The training proceeded for a fixed number of 500 epochs, and the parameters that yielded the highest validation accuracy were chosen.

## Data availability

All data used in this paper is available at neuroir.org and https://doi.org/10.5281/zenodo.13341219[96].

## Code availability

All code used in this paper is available at neuroir.org and https://doi.org/10.5281/zenodo.13341219[96].

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

## Acknowledgements

We owe our thanks to Giacomo Indiveri for his careful and detailed feedback. Yulia Sandarmiskaya and Jörg Conradt deserve our gratitude for their helpful comments. The authors would like to acknowledge the Telluride Neuromorphic Cognition Engineering Workshop for providing a conducive environment for conceiving and implementing the first steps towards NIR. The authors would like to acknowledge fundings from EC Horizon 2020 Framework Program under Grant Agreements 785907 and 945539 (J.E.P), the Pioneer Center for AI, under the Danish National Research Foundation grant number P1 (J.E.P), European Union's Horizon 2020 Research and Innovation Program under the Marie Skłodowska-Curie Grant Agreement No. 860360 (S.A.), the German Research Foundation as part of Germany's Excellence Strategy - EXC 2050/1 - Project ID 390696704 - Cluster of Excellence "Centre for Tactile Internet with Human-in-the-Loop" of Technische Universität Dresden (M.J.), the German Federal Ministry for Economic Affairs and Climate Action under the contract 01MN23004F (B.V.), the European Union - NextGenerationEU Project 3A-ITALY MICS Spoke 6, grants PE0000004, CUP E13C22001900001 (V.F.), the Italian National Recovery and Resilience Plan (NRRP), M4C2, funded by the European Union - NextGenerationEU, grant numbers IR0000011, CUP B51E22000150006, "EBRAINS-Italy" (G.U.), the ECSEL Joint Undertaking grant agreements 826655 "TEMPO" and 876925 "ANDANTE"; the KDT Joint Undertaking grant agreement 101097300 "EdgeAI"; Innosuisse and the Swiss State Secretariat for Education, Research and Innovation (D.R.M. and S. Sheik.), the Department of Energy's Office of Science contract DE-AC02-76SF00515 with SLAC through an Annual Operating Plan agreement WBS 2.1.0.86 from the Office of Energy Efficiency and Renewable Energy's Advanced Manufacturing and Materials Technology Office and the institutional support from SLAC National Laboratory (S.Shankar), the National Science Foundation under Award ECCS-2332166 (J.K.E.).

## Author contributions

J.E.P., S.A., M.J., F.C.B., B.V., G.L., and S. Sheik contributed to the NIR Core library. J.E.P., S.A., M.J., V.F., F.C.B., B.V., G.L., D.R.M., P.Z., K.H., T.C.S., S. Sheik, and J.K.E. contributed to the conversions and experiments for the different library and hardware backends. J.E.P., S.A., B.V., S. Sheik, G.U., V.F., and S. Shankar conceived the experiments. J.E.P., S.A., M.J., and J.K.E. contributed to the figures. All authors contributed to writing the paper.

## Funding

## Competing interests

The authors declare no competing interests.

## Additional information

[1]KTH Royal Institute of Technology, Stockholm, Sweden. [2]CogniGron Center, University of Groningen, Groningen, Netherlands. [3]Bernoulli Institute, University of Groningen, Groningen, Netherlands. [4]Technische Universität Dresden, Dresden, Germany. [5]Centre for Tactile Internet with Human-in-the-Loop, Dresden, Germany. [6]Neurobus, Toulouse, France. [7]Politecnico di Torino, Turin, Italy. [8]SynSense, Zurich, Switzerland. [9]LuxiTech Co. Ltd., Shenzhen, China. [10]University of Cambridge, Cambridge, UK. [11]Stanford University, Stanford, CA, USA. [12]SLAC National Laboratory, Menlo Park, CA, USA. [13]National Research Council, Ottawa, Canada. [14]University of California, Santa Cruz, USA. [15]These authors contributed equally: Jens E. Pedersen, Steven Abreu. ✉e-mail: jeped@kth.se

