## [Peer Review File · Nature Communications]

Neuromorphic intermediate representation: a unified instruction set for interoperable brain-inspired computingREVIEWER COMMENTS

Reviewer #1 (Remarks to the Author):

As stated by this article, now there are a plethora of neuromorphic software and hardware implementations with their own technology stacks, which challenges the reproducibility and reliability across platforms.

The problem this article is trying to address is very important; the authors have grasped it and came up with their proposal, Neuromorphic Intermediate Representation (NIR). However, this solution is too simple (from the point of view of computer systems, it can be said that the design idea is outdated) to solve the problem. And this flaw cannot be fixed with a few partial adjustment.

First, NIR is clearly influenced by the design methodology of compilation infrastructure such as LLVM for general-purpose processors (as shown in Figure 2 (e), (f); it is stated that instead 30 different compilers covering all $m \times n$ cases, only $m + n$ interfaces between each hardware and software platform to NIR are necessary). However, it should be pointed out that a single IR-based approach is not appropriate for the current era of domain-specific architecture as the main trend. Brain-inspired computing chips are not just a kind of DSA, as they come in a variety of architectures (thinking SpiNNaker, Loihi, TianjiC, and many other ReRAM-based chips, all of which have very different architectures). For DSA, there has been a significant shift in the compilation methodology away from relying on the design of single IR(s) and exploring more complex solution frameworks such as MLIR/TVM.

Second, as a deduction of the previous point, solutions like NIR are not fundamentally different from existing SNN-related intermediate representations (or description languages, such as PyNN, NeuroML, etc.), although NIR supports more hardware backends and gives some SNN examples.

Third, in terms of comparison with existing work, this article pits existing IRs against the representation of neuromorphic models. It emphasizes that existing representations (such as deep learning representations) are based on digital instructions in discrete time, while neuromorphic models had better been described as dynamical systems in continuous time. But I don't understand what this has to do with IR. Shouldn't a good IR be independent of the underlying hardware implementation? For example, MLIR is a reusable and extensible compiler infrastructure that can define IR (or dialects) at different levels of abstraction, which is not limited to conventional machine instruction set architectures.

This work makes a lot of sense. However, the proposal itself is simple and uninnovative, nor can it achieve its stated purpose, that is, a layer of abstraction can be established to achieve greater interoperability: This requires a lot more effort than an IR and its auxiliary work.

Reviewer #2 (Remarks to the Author):

Summary:

The paper describes a set of primitives which is commonly supported among neuromorphic hardware and simulators. Defining a neural network computational graph with these primitives enables scientists to train a network on one platform and export the model to other platforms. The cross-platform translation is tested on 3 different experiments: a simple spiking neural network instantiated with Norse, a recurrent spiking neural network trained with BPTT in snnTorch, and a convolutional spiking neural network. The quantitative results are either reported in the form of activity vector similarity or classification accuracy.

General opinion:

I am favorable to the publication of this paper although I see two major unknown to evaluate the impact or NIR depend: will there be a long-term commitment to maintain the code? will there be useful neuromorphic algorithms and applications relying on the NIR primitives?

Significance:

The cross-platform conversion software ONNX has shown to be very important for deep learning research and industry because it became a necessary gateway to convert pytorch deeplearning models for tensorflow. The authors are correctly mentioning this as a reference work. This comparison highlights the long-term code development and maintenance which is required in NIR as in ONNX to keep stable conversion code when framework versions are updated. The NIR Project is therefore a longterm commitment which might thrive if the community is strong, the main actors are committed and it becomes the unique way to bind popular neuromorphic software efficiently.

PyNN and NeuroML are also rightfully mentioned as reference work, in fact they do support recurrent spiking neural network and enable to from some spiking simulators to neuromorphic hardware. The conclusive comment from the authors: "Both PyNN and NeuroML focus on computational neuroscience and not on the scalable compute capabilities of SNNs" indicates that NIR is developed by a difference community but I find the term "scalable compute capabilities of SNNs" too vague to point at a precise technical feature of NIR which will make it stand against PyNN or NeuroML. If I try to be cynical and humorous, is NIR the 15th standard (see <https://imgs.xkcd.com/comics/standards.png>)?

Suggestions for improvement:

1) A major critique of neuromorphic computing beyond this paper is that neuromorphic applications are lacking behind deep learning and AI applications. Although this paper is not meant to resolve this problem, the significance of this work greatly depends on the speed of development of applications relying on the NIR primitives. For instance, some of the authors have recently published interesting algorithms like Spike-GPT which push the boundaries of neuromorphic algorithms, but they require algebraic operations (float matrix multiplications) which are outside of the standard NIR primitives as far as I understand it. So concretely, I miss a summary of the state of application using SNN, SRNN as available in NIR: what are the most advanced applications that are compatible with NIR: LIF ResNets? Audio Key-word detection SRNN (what about general speech-to-text translation) ? and How are those applications interesting for neuromorphic in comparison with deep learning application ? This is important to evaluate whether NIR based algorithms are already a deprecated technology or if there are cutting edge neuromorphic application with deep-learning-like capabilities relying on NIR? If the point is energy efficiency an argumentation of the potential of NIR primitives for energy efficiency with documented references is needed (and a comparison with energy efficient deep learning solutions would be greatly appreciated).

2) Since the NIR framework is oriented toward neuromorphic applications (in comparison with PyNN or NeuroML), I miss important details about the construction and the relevance of the chosen simulation experiments.

Which of the experiments has specifications which making it unadapted for PyNN and more adapted for NIR ? do you have numbers to support this argument ?

About Figure 4, b-d):

It is not clear to me how big in the network in tabe 1 and whether it is a random network or trained to relevant accuracy.

The calculation of the similarity percentage is not detailed: e.g. is the percentage corresponding to a relative L2 distance: $\|a - b\| / \max(\|a\|, \|b\|)$?

I hope that the calculation of activity similarity is correcting for the time step difference in Lava and small things like that which are likely to have no impact on a downstream function and can apparently be accounted for by displaced the spike/voltage vector by one time step.

There are obvious things that impact the strict activity equality like weight quantization, but this can be accounted for with quantization aware training on the initial network. Which experiments are tested with quantized weights? Would that avoid most of the activity mismatch?

About table 1:

Are the model trained with quantization aware training?

Missing a highlight of the source framework in which the network is trained or instantiated in the table. It is trivial that the accuracy is expected to be high in the source software or with similar frameworks.

The experiments could be described in more details: in each case, what is the training dataset? training to what accuracy? what is the network architecture? (I hope at least one of these experiment is large enough hit the "scale limits" of PyNN or NeurML)

Missing visual summaries: table of feature-compatibibility? Or grouping of frameworks?

On a related note about Figure 4: I miss a summary feature which explain the mismatch activity and accuracy (weight quantization? non-determinism? asynchronicity? approximation of some primitives?).

A related issue would be to highlight which framework is enable hardware acceleration: norse for instance is only implementable on CPU/GPUs (non neuromorphic primitive specific, does it have at least cuda-level routine?), spyx handles XLA compilation (also not neuromorphic primitive specific), xylo, Spinnaker 2 and lava software are bit-equal hardware models making them more adapted for milliWatt hardware compilation (more neuromorphic hardware specific). Those software are therefore not equal are could be grouped in corresponding categories in the quantitative summaries (Figure 4 and Table 1)

Reviewer #2 (Remarks on code availability):

The code is provided and is potentially an important part of the submission as it provides the long-term cross-platform interface to enable conversion across neuromorphic hardware.

Missing code samples:

A lot of the value of NIR is in the read-and-write routines to convert models from/to NIR. So far I only saw conversion codes for Norse and Nengo. More code was necessary to perform the comparison table in Figure 4 and Table 1. It would be great to publish the code that was used to produce the Figures (even if it has incomplete NIR primitive support for some platforms).

In my view publishing the training/conversion code for the three experiments would be easier and even more valuable than written specifications in the paper.

Reviewer #3 (Remarks to the Author):

In this paper, the authors establish a common reference-frame for computations in digital neuromorphic systems, entitled Neuromorphic Intermediate Representation (NIR). Computational model primitives are composed into graphs, which abstract information about discretization and hardware constraints. This allows for the representation of different models to be mapped to and from various neuromorphic technology stacks. The authors demonstrate this across seven neuromorphic simulators and four hardware platforms.

First, I would like to thank the authors for submitting their work and making their codes publicly accessible. I have the following specific concerns and comments for consideration:

1. The authors state "The field of deep learning faced similar incompatibility challenges a decade ago, but since then, intermediate representations and compiler frameworks such as ONNX [18], MLIR [19], XLA [20], and TVM [21] have bridged the gap between model description and different hardware accelerators". While these representations and tools have certainly aided the development of software stacks and deployment toolchains for various hardware platforms, they are quite fragmented and inflexible. For example, TVM is designed to parallelize operations within layers, and to execute neural network layers sequentially, and while MLIR dialects can be used to

represent high-level abstractions of neural network models, they are much better suited for local code generation. Many popular machine learning frameworks support exporting models to ONNX, however, they are unable to be imported (e.g., PyTorch). While these tools are all useful in different scenarios, the development of a unified representation/tool that can be used interchangeably between high-level frameworks and compilers is very much still an open problem, and arguably has not been “bridged”. In the context of neuromorphic systems, NIR is more akin to <https://github.com/microsoft/MMdnn> than these listed tools. NIR appears to facilitate the high-level and hardware-agnostic representation of neuromorphic models and enables integration between different neuromorphic simulator tools. It does not appear to bridge the gap between model description and different hardware accelerators, as claimed (note that the deployment of all digital neuromorphic hardware platforms is performed through neuromorphic simulator tools, except SpiNNaker2, for which networks can be defined using a high-level representation inspired by PyNN).

2. How exactly can the continuous nature of NIR be applied to mixed-signal hardware platforms?

3. The number of primitives is quite limited. It would be appreciated if the authors could discuss what network configurations are currently not supported and whether these would be considered in future? E.g., adaptive neuron thresholds and more complex neuron models. Will something akin to different opsets/releases be adopted?

4. More information about how constrains can be codified into a defined set of NIR graphs is needed. Could the authors provide a simple example of how this is done in a practical sense? In my opinion, the language used here is superfluous and should be changed – defining a check for specific properties of the graph for each tool/platform is a rather trivial process, and there is no need to codify these constrains into a defined set of graphs.

5. The following URL is invalid: <https://github.com/neuromorphs/NIR/tree/paper/paper>.

Reviewer #3 (Remarks on code availability):

The code is well structured and documented. I was able to use it to export one network from a tool to another.

Reviewer #1

As stated by this article, now there are a plethora of neuromorphic software and hardware implementations with their own technology stacks, which challenges the reproducibility and reliability across platforms.

We thank the reviewer for providing their thoughtful critique of NIR and recognizing the importance of increased reproducibility and interoperability in neuromorphic computing. We agree that as neuromorphic architectures continue to diversify, more sophisticated solutions will be needed for seamless interoperability. NIR is not positioned as the final answer, but rather as the seed for a unified neuromorphic representation—a proof of concept demonstrating reproducible functionality across 11 fragmented platforms. In a diversified landscape made of hardware and software that grow at different paces, NIR shines a light on the common substrate of primitives, showing how to harness them to move across platforms.

The problem this article is trying to address is very important; the authors have grasped it and came up with their proposal, Neuromorphic Intermediate Representation (NIR). However, this solution is too simple (from the point of view of computer systems, it can be said that the design idea is outdated) to solve the problem. And this flaw cannot be fixed with a few partial adjustment.

We appreciate that the reviewer points out the importance of the problem. Regarding the assertion that our Neuromorphic Intermediate Representation (NIR) proposal is too simplistic and outdated from a computer systems viewpoint, we argue that NIR is not meant as a wholesale solution to bridge systems or platforms. Rather, NIR is meant to translate multiple different representations to heterogeneous hardware. While we draw inspiration from traditional IR approaches like LLVM, NIR does not compile or translate any code and does not aim to substitute description languages or libraries. We built a lightweight representation tailored specifically for neuromorphic systems, which have fundamentally different computational models compared to general-purpose processors or conventional domain-specific architectures (DSAs). Instead of relying on discrete instructions, NIR models physics-friendly computational processes that, in the present paper, translates across 11 platforms. We also note that our approach does not preclude the use of other, (digital) intermediate representations and compilers.

First, NIR is clearly influenced by the design methodology of compilation infrastructure such as LLVM for general-purpose processors (as shown in Figure 2 (e), (f); it is stated that instead 30 different compilers covering all $m \times n$ cases, only $m + n$ interfaces between each hardware and software platform to NIR are necessary). However, it should be pointed out that a single IR-based approach is not appropriate for the current era of domain-specific architecture as the main trend. Brain-inspired computing chips are not just a kind of DSA, as they come in a variety of architectures (thinking SpiNNaker, Loihi, TianjiC, and many other ReRAM-based

chips, all of which have very different architectures). For DSA, there has been a significant shift in the compilation methodology away from relying on the design of single IR(s) and exploring more complex solution frameworks such as MLIR/TVM.

The reviewer makes fair points about limitations compared to multi-level compiler abstractions that are better suited for specialized hardware mappings.

1. Firstly, we argue that NIR targets a more foundational step than many contemporary cross-compilation frameworks for deep learning. Before optimizing performance, establishing a baseline for model reproducibility is needed, similar to model-centric APIs in deep learning. NIR currently focuses on the portability of core model dynamics rather than direct hardware efficiency and model performance, as modern and larger compiler frameworks for deep learning do, such as MLIR/TVM.
2. Secondly, our approach does not preclude the use of more complex, multi-level IR frameworks analogous to MLIR dialects in the future. NIR serves as an initial step towards establishing a common language for describing neuromorphic models, which can then be mapped to specific hardware backends. MLIR and its dialects were developed only once deep learning hardware and software have reached a level of maturity that neuromorphic computing is still advancing towards. In Figure 1, we illustrate that it is entirely possible to translate NIR to a time-discretized model, allowing us to integrate with digital dialects, if needed. Translating discrete models from digital IRs to continuous-time is possible, but will result in numerical imprecisions.

We have refined our claims around full-fledged compilation capabilities or solving interoperability in one step. Supporting incremental progress is still important, despite valid critiques about simplicity. The reviewer provides excellent suggestions for future NIR extension, integrating hardware details and toolchains. We added a discussion of limitations and future directions.

Changes to the manuscript:

- *Discussion of limitations and future directions*
- *Refined claims around compilation capabilities and solving interoperability*

Second, as a deduction of the previous point, solutions like NIR are not fundamentally different from existing SNN-related intermediate representations (or description languages, such as PyNN, NeuroML, etc.), although NIR supports more hardware backends and gives some SNN examples.

We argue that NIR distinguishes itself from existing SNN representations like PyNN or NeuroML in at least three ways:

1. NIR is a declarative abstraction consisting solely of implementation-independent primitives. PyNN and NeuroML (and Fugu and Lava) are indeed describing abstract computational models on some level, as the reviewer points out, but their computational

models do not stand alone. PyNN is closer to a programmatic frontend and cannot be serialized. Hence, the PyNN models are closely tied to PyNN as a framework and models require platform-specific Python code to run by setting and configuring the specific PyNN backend. NeuroML defines XML-based computational models that are independent of their implementation, like NIR, but is dedicated to describing generic and customizable “biophysics, anatomy and network architecture of neuronal systems at multiple scales” (from <https://docs.neuroml.org/Userdocs/MissionAndAims.html>). By focusing on biophysics and anatomy, NeuroML uses primitives at the neuronal level. E.g., an Izhikevich neuron in NeuroML is made to map across to an Izhikevich neuron in PyNN. NIR sits at a lower abstraction level in that continuous-time ODEs are treated as the computational primitives, rather than neuron models. To make NIR exchange models between different libraries, the ODEs/primitives are chained together to construct neuron models. As such, we avoid focusing on any particular domain, like biophysics and anatomy.

2. Isolating the primitives as described above makes NIR compatible with existing ML frameworks, such that executable models can be transferred between systems and libraries. A PyNN model in the form of a Python script is not platform-independent because code needs to change before it works on another platform.
3. PyNN and NeuroML focus on computational neuroscience models, while NIR identifies simpler hybrid systems primitives. One important sub-point is that we obtain higher-order differential equations by composing NIR primitives. This is possible, but hard to do, in other representations.

Finally, NIR provides a level of agreement and compatibility between many platforms from industry and academia that is unprecedented in neuromorphic computing, and provides a unique fertile basis for further work that we outlined above. We added a section 2.4 in the paper which more directly describes the relations to other intermediate representations in a form that, we believe, is more accessible to the reader.

Changes to the manuscript:

- *Reworked introduction and related work sections to (1) clarify the aim and scope of the paper and (2) contrast our work with existing efforts*
- *Added section 2.4 in the paper describing the relation to other intermediate representations*
- *Added table comparison of ONNX, MLIR, LLVM, PyNN, and NeuroML in Supplementary Material, section B*

Third, in terms of comparison with existing work, this article pits existing IRs against the representation of neuromorphic models. It emphasizes that existing representations (such as deep learning representations) are based on digital instructions in discrete time, while neuromorphic models had better been described as dynamical systems in continuous time. But I don't understand what this has to do with IR. Shouldn't a good IR be independent of the underlying hardware implementation? For example, MLIR is a reusable and extensible compiler

infrastructure that can define IR (or dialects) at different levels of abstraction, which is not limited to conventional machine instruction set architectures.

We agree with the reviewer that an IR should be hardware-agnostic, in the sense that the representation does not make use of instructions/primitives that are only available on some hardware platforms. The difficulty with MLIR and other digital solutions, is that they are incapable of representing analog computations and thus are not suitable for mixed-signal and analog hardware. This is precisely why we need NIR. By using a continuous-time formulation, NIR is more general than any digital IR because NIR does not assume the existence of digital instruction sets.

We would like to also point out that while the *representation* of the computation should be hardware-agnostic (in the sense explained above), the *execution* of the computation is always hardware-dependent. Even on conventional digital machines, a simple operation like the dot product can deviate numerically when executed sequentially on a CPU vs. in parallel on a GPU (because of differences in floating-point rounding errors due to the different execution order). We believe a core contribution of NIR is to show that such execution differences lead to significant mismatch for neuromorphic computations, even across digital platforms (device mismatch is already well-known as a problem on analog platforms), and that such mismatches are worse for some computations (our SRNN example) than others (our SCNN example). We believe that NIR brings an additional layer to the computing stack as an extra degree of freedom and that it provides a unique tool for the community to investigate and fix mismatches.

We further agree that tools like MLIR are not limited to conventional machine instruction set architectures. E.g., MLIR has been applied to near-memory and in-memory computing (<https://doi.org/10.48550/arXiv.2301.07486>). However, what one can define with MLIR is a sequence of operations, but not a continuous-time system. Thus there is a need for the neuromorphic intermediate representation to fill this gap. Note that a similar problem applies to the Fourier transform: while there is no abstraction of the Fourier transform in its integral form in MLIR, one can have a dialect for the discrete Fourier transform or FFT algorithms in MLIR.

MLIR and LLVM have much value, but fit a different need. We envision a possible future where NIR could be used in tandem with compiler frameworks, such that MLIR/LLVM model discrete operators and connections, while NIR introduces compatibility with continuous-time functions. NIR itself may in the future also offer further levels of abstractions (or dialects) for optimization, e.g. for the fusion of operations or the graph flattening.

This work makes a lot of sense. However, the proposal itself is simple and uninnovative, nor can it achieve its stated purpose, that is, a layer of abstraction can be established to achieve greater interoperability: This requires a lot more effort than an IR and its auxiliary work.

We agree that establishing true cross-platform interoperability requires significant effort beyond just defining an IR. We further agree that NIR itself is simple, but emphasize that the efforts to get NIR working across all the various platforms are far from trivial. We, therefore, argue that

NIR is an innovative and crucial first step toward physical computation that lays the foundation for an open, vendor-neutral ecosystem for neuromorphic computing.

We respectfully disagree with Reviewer 1's judgment of NIR as "simple and uninnovative" [sic] and unable to "achieve its stated purpose". We believe that the experiments from our paper clearly demonstrate that we can indeed achieve our stated purpose of providing "a layer of abstraction" to "achieve greater interoperability". We kindly refer to the other reviewers, acknowledging that NIR indeed "facilitates the high-level and hardware-agnostic representation of neuromorphic models".

That being said, wide early adoption of this representation shows value in a common model definition as more complex solutions are developed. Only with significant collaboration among groups like the reviewer's can neuromorphic computing advance. We would welcome the chance to work together to address these challenges.

Rebuttal #2

Summary:

The paper describes a set of primitives which is commonly supported among neuromorphic hardware and simulators. Defining a neural network computational graph with these primitives enables scientists to train a network on one platform and export the model to other platforms. The cross-platform translation is tested on 3 different experiments: a simple spiking neural network instantiated with Norse, a recurrent spiking neural network trained with BPTT in snnTorch, and a convolutional spiking neural network. The quantitative results are either reported in the form of activity vector similarity or classification accuracy.

We are deeply grateful for the reviewer's detailed and constructive feedback. We proceed to address the review point-by-point below.

General opinion:

I am favorable to the publication of this paper although I see two major unknown to evaluate the impact or NIR depend: will there be a long-term commitment to maintain the code? will there be useful neuromorphic algorithms and applications relying on the NIR primitives?

We fully agree that for NIR to have a lasting impact, there needs to be a strong community commitment to actively maintaining and updating the code over time, similar to efforts like ONNX. We have built an active community around NIR through open development on GitHub and have already started building collaborations with several academic and industrial partners. Many of the authors are dedicated to maintaining and extending NIR compatibility with their software and hardware platforms, and multiple academic projects are ongoing. Several authors maintain widespread libraries that guarantee some degree of maintenance, and the open-source

model also invites further external contributions. We are further working on other sustainable organizational settings.

Regarding cutting-edge applications compatible with NIR, we have clarified that NIR focuses on scalable spiking neural networks rather than biological modeling. NIR can represent complex neural architectures like LIF ResNets and SRNNs for audio sensing, and we cited recent work showing SRNNs matching keyword spotting accuracy of LSTM networks. Thanks to the reviewer, we have created a NIR roadmap to communicate the concrete plans we have for NIR, available at <https://neuroir.org/docs/roadmap.html>. Those include ways to extend NIR to represent models like SpikeGPT, additional neuron models like the Fitzhugh-Nagumo and Izhikevich models, and support for libraries like lava-dl, lava-dnf, lava-optimization, Nengo, as well as Fugu. We additionally added references demonstrating state-of-the-art neuromorphic algorithms and their energy efficiency benefits compared to deep learning.

Changes to the manuscript:

- *Improved documentation and code for the network descriptions and training in the code for increased maintainability and reproducibility*
- *Added a roadmap available at <https://neuroir.org/docs/roadmap.html>*

Significance:

The cross-platform conversion software ONNX has shown to be very important for deep learning research and industry because it became a necessary gateway to convert pytorch deeplearning models for tensorflow. The authors are correctly mentioning this as a reference work. This comparison highlights the long-term code development and maintenance which is required in NIR as in ONNX to keep stable conversion code when framework versions are updated. The NIR Project is therefore a longterm commitment which might thrive if the community is strong, the main actors are committed and it becomes the unique way to bind popular neuromorphic software efficiently.

We fully agree, with NIR we are committed to finding a long-term strategy to bind neuromorphic platforms efficiently and robustly, see our comment above.

PyNN and NeuroML are also rightfully mentioned as reference work, in fact they do support recurrent spiking neural network and enable to from some spiking simulators to neuromorphic hardware. The conclusive comment from the authors: "Both PyNN and NeuroML focus on computational neuroscience and not on the scalable compute capabilities of SNNs" indicates that NIR is developed by a difference community but I find the term "scalable compute capabilities of SNNs" too vague to point at a precise technical feature of NIR which will make it stand against PyNN or NeuroML. If I try to be cynical and humourous, is NIR the 15th standard (see <https://imgs.xkcd.com/comics/standards.png>)?

This sentiment is echoed by Reviewer 1 who highlights the importance of addressing the distinction between NIR and NeuroML/PyNN more clearly. In our reply to reviewer 1, we mention

that NIR distinguishes itself in three ways that we have not seen before in the literature. First, NIR is only providing computational primitives, not any execution-related details, such as backend configuration etc., as we see in PyNN in particular. Second, NIR is designed to coexist with other standards and compiler toolkits, for instance from the ML/DL world, exactly to avoid NIR operating in isolation from other works and standards. Third, NIR is operating at a different level of complexity, which is where the “scalable compute capabilities of SNNs” from the initial draft comes into play. We have additionally added a section 2.4 in the paper describing this in the paper to clarify exactly what technical features distinguish NIR from previous work.

Changes to the manuscript:

- *Reworked introduction and related work sections to (1) clarify the aim and scope of the paper and (2) contrast our work with existing efforts. Specifically, we removed claim about “scalable compute capabilities in SNNs”. Instead, we refer to a temporal scaling component in section 2.4 (see bullet point below) and large-scale systems which we touch upon in the discussion*
- *Added section 2.4 in the paper describing the relation to other intermediate representations*
- *Added table comparison of ONNX, MLIR, LLVM, PyNN, and NeuroML in Supplementary Material, section B*

Suggestions for improvement:

1) A major critique of neuromorphic computing beyond this paper is that neuromorphic applications are lacking behind deep learning and AI applications. Although this paper is not meant to resolve this problem, the significance of this work greatly depends on the speed of development of applications relying on the NIR primitives. For instance, some of the authors have recently published interesting algorithms like Spike-GPT which push the boundaries of neuromorphic algorithms, but they require algebraic operations (float matrix multiplications) which are outside of the standard NIR primitives as far as I understand it. So concretely, I miss a summary of the state of application using SNN, SRNN as available in NIR: what are the most advanced applications that are compatible with NIR: LIF ResNets? Audio Key-word detection SRNN (what about general speech-to-text translation) ? and How are those applications interesting for neuromorphic in comparison with deep learning application ? This is important to evaluate whether NIR based algorithms are already a deprecated technology or if there are cutting edge neuromorphic application with deep-learning-like capabilities relying on NIR? If the point is energy efficiency an argumentation of the potential of NIR primitives for energy efficiency with documented references is needed (and a comparison with energy efficient deep learning solutions would be greatly appreciated).

Spiking neural networks lag behind artificial neural networks in terms of the scale of tasks they achieve. One of several reasons is due to the hardware lottery: GPUs are accessible, and modern deep learning has grown to exploit GPUs. Neuromorphic hardware is limited in accessibility, and necessarily lag behind as a result. We strongly believe that tools like this should not be viewed as "dependent on the speed of development of applications", but rather, can accelerate the

speed of development of applications. While we are hopeful that a tool like this will lead to a demonstration of the "neuromorphic advantage", it will nonetheless accelerate the process of determining whether such an advantage can exist. Whether it is a positive or negative result, NIR can accelerate the discovery of both and reduce resource-waste by speeding up time-to-deployment.

We agree that it would be helpful to clarify the state-of-the-art performance for network architectures supported by NIR. We would also like to point out that there are open discussions in the literature pointing to methodological problems in neuromorphics; it is currently challenging to provide fair and reproducible comparisons, exactly because the technology stacks and neuron models are so vastly different. The NeuroBench initiative is one possible candidate towards more robust and correct baseline comparisons.

That said, we addressed the reviewer's comment by adding a paragraph to the Discussion about the limitations of the shown experiments, a brief overview of SOTA SNN architectures, and references about energy efficiency comparison between neuromorphic and standard hardware. More concretely, we provide references to existing work within LIF Resnets, SCNNs, SRNNs, and spiking transformer-inspired architectures, such as SpikeGPT ([arxiv:2302.13939](https://arxiv.org/abs/2302.13939)) and SpikFormer ([arxiv:2401.02020](https://arxiv.org/abs/2401.02020)). In addition we point to studies that compare the energy and latency of neuromorphic hardware solutions to conventional hardware (GPUs). All these works were published in the last few years. Given the support in NIR, or planned support in our roadmap, along with the intense publication cadence for the above-mentioned models, we argue that NIR will remain highly active for at least the foreseeable future.

Changes to the manuscript

- *Added section 2.1.4 on experimental mismatches and section 4.2 on hardware constraints*
- *Added discussion points about energy and large-scale neuromorphic systems*
- *Added references to energy efficiency in the introduction*

2) Since the NIR framework is oriented toward neuromorphic applications (in comparison with PyNN or NeuroML), I miss important details about the construction and the relevance of the chosen simulation experiments.

We intended to cover a representative set of neuromorphic network architectures and computational primitives. Specifically, we argue that:

1. The single LIF neuron experiment illustrates NIR's ability to capture fundamental, low-level spiking neuron dynamics, which is a core requirement for more complex neuromorphic computations.
2. The spiking convolutional neural network (SCNN) represents a commonly used architecture for pattern recognition problems like image classification, which is an important use case for neuromorphic computing.

3. The recurrent spiking neural network (SRNN) is relevant as RNNs are widely used in neuromorphic applications like speech recognition, control, and time-series classification or forecasting tasks.

We chose the well-known N-MNIST benchmark for the SCNN as readers will be very familiar with it already, and it is used across the literature as a baseline performance test.

For the SRNN, we decided on the Braille reading task, which introduces not just the requirement for temporal capabilities but also the ability to deal with spatio-temporal information. To fit the constraints of our smallest supported HW platform (the Xylo chip), we reduced the dataset to a subset ('A', 'E', 'I', 'O', 'U', 'Y' and 'Space') of the originally proposed characters. Other well-known tasks would not have worked (such as the Spiking Heidelberg Digits which as it has over 700 input channels), given the Xylo chip has only 16 input channels. We have added this, and more, information about the experiments to the paper.

Changes to the manuscript:

- We added more details about the implementation and construction of the simulation experiments to Sections 2.1 and 4.4, as well as the motivation for our chosen experiments.

Which of the experiments has specifications which making it unadapted for PyNN and more adapted for NIR ? do you have numbers to support this argument ?

The single LIF neuron and the SRNN are compatible with PyNN but the SCNN is not compatible with PyNN, as it does not support Convolution-Connectors. Therefore, the SCNN would lead to a tremendous number of synapses using PyNN. E.g., consider the 2nd Conv2D layer in the SCNN (see https://github.com/neuromorphs/NIR/tree/main/paper/02_cnn#model-definition): In NIR only 1152 weights are defined while one would need to create 373 248 synapses in PyNN. While PyNN simulations with millions of synapses are technically no problem, we would sacrifice hardware efficiency when using PyNN for the network definition: As the neuromorphic hardware systems considered for the SCNN (Speck, Loihi 2 and SpiNNaker2) provide a "native" support for convolutions, using PyNN as is would not allow for use of this feature.

Finally, we would like to emphasize again that NIR is not meant to *replace* existing tools like PyNN, but rather to *complement* them. In principle, it is entirely possible to translate the NIR representation into PyNN, given that the NIR graph does not contain primitives that PyNN does not support.

About Figure 4, b-d):

It is not clear to me how big in the network in table 1 and whether it is a random network or trained to relevant accuracy.

We have included further details about the training setup of the SCNN and SRNN in Table 1 in the main text, and provide all details needed to replicate the training in a separate section (4.4).

The calculation of the similarity percentage is not detailed: e.g. is the percentage corresponding to a relative L2 distance: $\|a - b\| / \max(\|a\|, \|b\|)$?

Further detail has been added, and the exact equation that we used is described in the paper (section 2.1.2).

I hope that the calculation of activity similarity is correcting for the time step difference in Lava and small things like that which are likely to have no impact on a downstream function and can apparently be accounted for by displaced the spike/voltage vector by one time step.

The activity similarity is computed based on the spike rates, not individual spike timing, so no corrections are needed. We added clarifications for this in the paper.

There are obvious things that impact the strict activity equality like weight quantization, but this can be accounted for with quantization aware training on the initial network. Which experiments are tested with quantized weights? Would that avoid most of the activity mismatch?

All hardware platforms (Loihi, SpiNNaker2, Xylo, Speck) use quantized weights, which is also supported by the associated software platforms (Lava & Lava-dl, SpiNNaker2 simulations, Rockpool, and Sinabs, respectively).

We did not explore quantization-aware training techniques in our experiments and instead trained a full-precision network that is then quantized post-training, as quantization is only required upon deployment to a hardware platform. As highlighted by Reviewer 1, "a good IR is independent of the underlying hardware implementation". Therefore, any activity mismatch in software is independent of quantization and rather, a result of small variations such as sequence of operations (e.g., state update followed by spike emission followed by reset mechanism).

While the goal of the results are to show (1) that networks can run across devices, and (2) the degree of variation between platforms, quantization-aware training is a natural next step, but a fair treatment of QAT in the present paper requires quantization-aware training for each of the platforms.

We did not explore quantization-aware training techniques in our experiments, as quantization is only required upon deployment to a hardware platform. As highlighted by Reviewer 1, "a good IR is independent of the underlying hardware implementation". Therefore, any activity mismatch in software is independent of quantization and rather, a result of small variations such as sequence of operations (e.g., state update followed by spike emission followed by reset mechanism).

While the goal of the results are to show (1) that networks can run across devices, and (2) the degree of variation between platforms, quantization-aware training is a natural next step, but a

fair treatment of QAT in the present paper requires quantization-aware training for each of the platforms.

About table 1:

Are the model trained with quantization aware training?

No, they are not. The various hardware platforms have different constraints in bit-widths (across both weights and states), so a full precision baseline serves to isolate the effect of porting models across frameworks from hardware-dependent factors, such as quantizing models. In future work, we plan to explore quantization-aware training to improve cross-platform similarity, though we would like to emphasize that hardware constraints should be limited in assessing intermediate representations.

Missing a highlight of the source framework in which the network is trained or instantiated in the table. It is trivial that the accuracy is expected to be high in the source software or with similar frameworks.

We added section 4.2 to clarify some of the hardware constraints for training, which is the underlying cause for the missing frameworks in the tables. We further added details on which platform the networks were initially trained, as well as details about the training itself in section 4.4.

The experiments could be described in more details: in each case, what is the training dataset? training to what accuracy? what is the network architecture? (I hope at least one of these experiment is large enough hit the "scale limits" of PyNN or NeuroML)

As already pointed out above, the CNN is already outside of PyNN's compatibility: Using PyNN for defining the CNN would lead to generating a tremendous amount of synapses for convolution layers, which would also hinder efficient implementation on neuromorphic systems. The same applies to NeuroML2 and the LEMS simulator, which focuses on the simulation of more detailed neuron models and individual synaptic dynamics.

Changes to the manuscript

- *Added information about the training setup and the network architecture necessary to fully replicate our results in section 4.4.*
 - *We also uploaded the original training code to the GitHub repository, available at <https://github.com/neuromorphs/NIR/tree/main/paper>*
- *Added table comparison of ONNX, MLIR, LLVM, PyNN, and NeuroML in Supplementary Material, section B*

Missing visual summaries: table of feature-compatibility? Or grouping of frameworks?

We have decided not to include a table that showcases the feature-compatibility of these platforms, primarily because the software frameworks continue to evolve and compatibility will

change rather quickly, making our paper outdated or even misleading. For example, SpiNNaker2 can theoretically support any computation, which makes it very difficult (and not very meaningful) to make claims about what features SpiNNaker2 supports at the time of writing.

However, we agree that a grouping of frameworks can better guide the reader towards understanding the purpose and intent of the variety of frameworks. As such, we have grouped all supported platforms into two classes: those that are software-only, and those that run on hardware (possibly also offering a bit-accurate software simulator).

Additionally, we have added a comparison of ONNX, MLIR, LLVM, PyNN, and NeuroML based on their scope, language, supported language, and typical use case. The intention is to show illustrate the differences between the existing approaches while illustrating feature-compatibility on a higher abstraction level.

Changes to the manuscript

- *Added table comparison of ONNX, MLIR, LLVM, PyNN, and NeuroML in Supplementary Material, section B*

On a related note about Figure 4:

I miss a summary feature which explain the mismatch activity and accuracy (weight quantization? non-determinism? asynchronicity? approximation of some primitives?).

We thank the reviewer for pointing out this important gap in our paper. We have extended the paper with an entirely new section 2.1.4 to discuss the various causes for the mismatch in activity and accuracy. We also provide further explanations in the discussion section, with pointers to future work.

A related issue would be to highlight which framework is enable hardware acceleration: norse for instance is only implementable on CPU/GPUs (non neuromorphic primitive specific, does it have at least cuda-level routine?), spyx handles XLA compilation (also not neuromorphic primitive specific), xylo, Spinnaker 2 and lava software are bit-equal hardware models making them more adapted for milliWatt hardware compilation (more neuromorphic hardware specific). Those software are therefore not equal are could be grouped in corresponding categories in the quantitative summaries (Figure 4 and Table 1)

Related to the previous comment, we have now grouped the frameworks into those that are simulation-only, and those that (also) offer hardware acceleration. This nicely separates frameworks that work on general purpose hardware (CPU & GPU) from those that also support neuromorphic hardware (i.e., Loihi, Xylo, Speck and SpiNNaker2). We did not further separate spyx from Norse/snnTorch since both JAX and PyTorch are similar in their supported backends (CPU and GPU).

Reviewer #2 (Remarks on code availability):

The code is provided and is potentially an important part of the submission as it provides the long-term cross-platform interface to enable conversion across neuromorphic hardware.

Missing code samples:

A lot of the value of NIR is in the read-and-write routines to convert models from/to NIR. So far I only saw conversion codes for Norse and Nengo. More code was necessary to perform the comparison table in Figure 4 and Table 1. It would be great to publish the code that was used to produce the Figures (even if it has incomplete NIR primitive support for some platforms).

We absolutely agree. The code has been updated and is published on github.com/neuromorphs/nir/tree/main/paper. The README in the linked folders on GitHub contains more information about where to find the code to replicate our figures and tables, as well as code for how the different networks were read from (and written to) NIR.

We also updated the examples in the documentation, available at <https://neuroir.org/docs/>.

In my view publishing the training/conversion code for the three experiments would be easier and even more valuable than written specifications in the paper.

This is now cleaned up and available on the GitHub repo linked above.

Finally, we'd like to extend our appreciation to the reviewer for providing an external perspective which has helped the manuscript improve in clarity across the experimental setup, along with providing clear distinctions from other tools that are available.

Reviewer #3

In this paper, the authors establish a common reference-frame for computations in digital neuromorphic systems, entitled Neuromorphic Intermediate Representation (NIR).

Computational model primitives are composed into graphs, which abstract information about discretization and hardware constraints. This allows for the representation of different models to be mapped to and from various neuromorphic technology stacks. The authors demonstrate this across seven neuromorphic simulators and four hardware platforms.

First, I would like to thank the authors for submitting their work and making their codes publicly accessible. I have the following specific concerns and comments for consideration:

We thank reviewer #3 for their insightful comments. We proceed to address the reviewer's comments and concerns point-by-point.

1. The authors state “The field of deep learning faced similar incompatibility challenges a decade ago, but since then, intermediate representations and compiler frameworks such as ONNX [18], MLIR [19], XLA [20], and TVM [21] have bridged the gap between model description and different hardware accelerators”. While these representations and tools have certainly aided the development of software stacks and deployment toolchains for various hardware platforms, they are quite fragmented and inflexible. For example, TVM is designed to parallelize operations within layers, and to execute neural network layers sequentially, and while MLIR dialects can be used to represent high-level abstractions of neural network models, they are much better suited for local code generation. Many popular machine learning frameworks support exporting models to ONNX, however, they are unable to be imported (e.g., PyTorch). While these tools are all useful in different scenarios, the development of a unified representation/tool that can be used interchangeably between high-level frameworks and compilers is very much still an open problem, and arguably has not been “bridged”. In the context of neuromorphic systems, NIR is more akin to <https://github.com/microsoft/MMdnn> than these listed tools. NIR appears to facilitate the high-level and hardware-agnostic representation of neuromorphic models and enables integration between different neuromorphic simulator tools. It does not appear to bridge the gap between model description and different hardware accelerators, as claimed (note that the deployment of all digital neuromorphic hardware platforms is performed through neuromorphic simulator tools, except SpiNNaker2, for which networks can be defined using a high-level representation inspired by PyNN).

We agree that existing intermediate representations (IRs) for deep learning still face significant challenges, and a truly unified representation remains an open problem. We have moderated and clarified our claims regarding deep learning IRs bridging the gap completely. As the reviewer points out, NIR is more analogous to model-centric IRs like MMDNN or ONNX rather than layer-centric IRs like TVM. We clarify that NIR does not aim to map models directly to hardware, but rather serves as a common interface between different neuromorphic software and hardware stacks. In that sense, NIR does not in itself bridge any gap. Instead, it is a common representation between neuromorphic technology stacks that can serve as a tool for translating between A and B. We agree that it is incorrect to state that NIR bridges the gap between *model description and hardware accelerators*. We do, however, consider it correct to say that NIR *can* bridge software and hardware platforms that support the representation in two ways. Firstly, it can serve as a “lingua franca” for hardware platforms such that they do not have to implement their own simulation or modeling software. Secondly, it allows a single model to be run on multiple (supported) platforms. Both aspects are crucial when dealing with the incompatibility challenge mentioned above.

2. How exactly can the continuous nature of NIR be applied to mixed-signal hardware platforms?

In a mixed-signal neuromorphic system, analog signals are inherently continuous-time, and can be used to replicate the continuous dynamics of neural and synaptic models as described in

NIR. This includes the implementation of elements like leaky integrators, threshold mechanisms, and synaptic transmission processes. That means that NIR primitives could be realized as physical circuit components, and that the system is simulated as a whole rather than discretized. The continuous nature of NIR then provides the possibility of describing the behavior of the analog system more accurately, which will, in turn, help with standardization and deployment of different models. We have added text discussing this potential direction.

3. The number of primitives is quite limited. It would be appreciated if the authors could discuss what network configurations are currently not supported and whether these would be considered in future? E.g., adaptive neuron thresholds and more complex neuron models. Will something akin to different opsets/releases be adopted?

The reviewer raises an excellent point. We have added a discussion on current limitations of NIR primitives and plans to support more complex models like adaptive thresholds and multi-compartment neurons in future releases (c.f. the roadmap in response to reviewer 2). The opset model for defining versions is promising and has been added to the discussion.

Currently, we do not support models that can *not* be composed by leaky integrators and spike-threshold functions. This includes quadratic models like the Izhikevich neuron model and cubic models like the Fitzhugh-Nagumo. But we can support complex conductance-based models, such as the Hodgkin-Huxley model, because NIR supports the composition of multiple leaky ion channels in parallel. It also includes multicompartmental models, which can be described as cable equations adjacent to a soma. Adaptation and plasticity are currently not yet supported. We list all these restrictions in the discussion.

For future consideration, the continuous-time dynamical systems in NIR are inherently compatible with analog computation, and the translation to physical analog components is an exciting next step. We see this as a crucial addition in supporting a wide variety of prospective neuromorphic hardware. One important part of this will need to involve a detailed exploration of how NIR's computational primitives can be physically instantiated in analog neuromorphic circuits.

4. More information about how constraints can be codified into a defined set of NIR graphs is needed. Could the authors provide a simple example of how this is done in a practical sense? In my opinion, the language used here is superfluous and should be changed – defining a check for specific properties of the graph for each tool/platform is a rather trivial process, and there is no need to codify these constraints into a defined set of graphs.

We agree the original text was unclear. We have rewritten this section to simply state that each hardware platform imposes constraints on executable NIR graphs, which can be checked before mapping a graph to hardware. We provide Xylo as a concrete example of such constraints.

However, we would like to argue that the involved graph matching is not a trivial process, it even becomes intractable when working with large graphs. It may be trivial when constrained to linear

or convolutional layers, but NIR is designed to handle arbitrary connections with feedback loops. The challenge is that the same computational graph may be matched with different sub-graph partitions – thus leading to many possible solutions that have to be tested.

Mathematically, this involves the subgraph isomorphism problem, which is known to be NP-complete. In a concrete example, consider a computational graph in NIR that we want to map to a specific hardware. The hardware offers a set of computational primitives that can be interconnected in some constrained way (e.g. limited fan-in). Finding subgraphs of the original NIR graph that are isomorphic to the computational primitive of our hardware, *can* be a simple one-to-one mapping if the HW primitives are directly supported by NIR, but the mapping may also be more complicated.

We ran into a simple example of this already in our experiments. Some frameworks represent recurrently connected LIF populations as a separate building block (e.g. the RLeaky neuron in snnTorch) while others represent it as a standard LIF population that is connected to itself (e.g. Nengo). The latter is the approach taken in NIR. Therefore, to parse an RNN from NIR into frameworks like snnTorch, we have to detect subgraphs that are isomorphic to **LIF<->Dense**. These are admittedly simple subgraphs, but this example hopefully helps illustrate the presence of the general subgraph isomorphism problem for more complex scenarios.

We added details on this in sections 4.2 and 4.3 to highlight this problem both theoretically and practically.

5. The following URL is invalid: <https://github.com/neuromorphs/NIR/tree/paper/paper>.

This has been fixed in the revised manuscript. The code is available at this URL: <https://github.com/neuromorphs/NIR/tree/main/paper>

Reviewer #3 (Remarks on code availability)

The code is well structured and documented. I was able to use it to export one network from a tool to another.

We are delighted to hear that our code is easy to use. We will continue to strive for reproducibility and openness.

REVIEWER COMMENTS

Reviewer #1 (Remarks to the Author):

I would like to thank the authors for their detailed responses to the reviewers' comments very much, and I acknowledge the authors' descriptions of the uniqueness and role of NIR, although there are still reservations about its importance.

Building a neuromorphic ecosystem is urgently needed and, as the authors say, requires the full cooperation of the entire community. But I don't think the NIR is an important task or a critical starting point, although it is useful; The responses also did not provide sufficient arguments to justify their importance. The details are as follows:

1. First of all, the authors emphasize that they draw inspiration from traditional IR approaches. But I don't think authors took the most crucial inspiration from the traditional systems. The core contribution of traditional compilation infrastructures such as LLVM and the newer MLIR does not lie in the two-layer or multi-layer IRs: In the case of LLVM, the key contribution is to build many reusable optimization conversion codes (including front-end optimization and back-end optimization) around the upper-layer IR and the lower-level IR respectively, to reduce the difficulty and effort of building compilers for new processors. IR itself is not very critical. This holds true for MLIR, too. That's why LLVM and MLIR are called compiler infrastructures.

2. Authors seem to think that current compilation infrastructures such as MLIR cannot represent continuous-time dynamical systems. I disagree with that. The root of MLIR's design advantage lies not in its existing multiple dialects, but in the dialect mechanism itself, which can be flexibly extended and descended layer by layer. NIR may serve as a high-level dialect in MLIR. These two can't be compared directly, and as mentioned above, the contribution of NIR has nothing to do with the key of MLIR.

3. The authors' work focuses on a single-layer IR and highlights the differences from existing IRs such as PYNN. But is it necessary to introduce a new intermediate representation for being able to represent continuous-time dynamical systems? Does this in itself exacerbate fragmentation? This can be achieved with some extensions to existing IR or programming languages.

Reviewer #2 (Remarks to the Author):

Thank you for addressing my review.

In general, I believe the field would benefit from moving away from this artificial frontier of those mathematical abstractions. The important and hard question for the field is whether any of these models and hardware are better than other systems (deep learning with custom accelerators). This theoretical question is not solved by the model presented here, and there is the danger that the efforts put into the homogenization of the field around models which under-perform on benchmarks (in comparison to machine learning models) would slow the field down rather than bringing us forward toward successful neuromorphic mathematical models.

On the other hand, there is an important need to compare experimental hardware. The interoperable software being developed here might be a crucial milestone towards a systematic comparison of recent neuromorphic hardware which was designed for the previous generation of neuromorphic models. In this sense, the work is important to enable a careful comparison of contemporary simulation substrates with a fixed model. Unfortunately, the most valuable comparison results are not fully studied here (resource efficiency, scaling capability).

My conclusion is that, although the work is worth being published as a step toward a longer-term achievement. I am doubtful that the revised manuscript will be impactful as such.

Reviewer #3 (Remarks to the Author):

I would like to thank the authors for carefully responding to and addressing my comments/concerns. In my opinion, the paper is now suitable for publication in Nature Communications. I would also like to thank the authors for their ambitious vision and clearly documenting and making their codes available - developing open source and community orientated tools is often a thankless effort.

Reviews July 2024

Reviewer #1 (Remarks to the Author):

I would like to thank the authors for their detailed responses to the reviewers' comments very much, and I acknowledge the authors' descriptions of the uniqueness and role of NIR, although there are still reservations about its importance.

Building a neuromorphic ecosystem is urgently needed and, as the authors say, requires the full cooperation of the entire community. But I don't think the NIR is an important task or a critical starting point, although it is useful; The responses also did not provide sufficient arguments to justify their importance.

We thank again the reviewer for their in-depth comments, which we are convinced have improved the quality of our submission, and for their acknowledgement of the difficulty of the task. We also appreciate their remarks about the need for better neuromorphic ecosystems. It is our hope to convince the reviewer that our contribution is a step in that direction. For reference, the preprint posted in arXiv has already been referenced 8 times since November 2023, an indication of the relevance of this work to the community.

The details are as follows: 1. First of all, the authors emphasize that they draw inspiration from traditional IR approaches. But I don't think authors took the most crucial inspiration from the traditional systems. The core contribution of traditional compilation infrastructures such as LLVM and the newer MLIR does not lie in the two-layer or multi-layer IRs: In the case of LLVM, the key contribution is to build many reusable optimization conversion codes (including front-end optimization and back-end optimization) around the upper-layer IR and the lower-level IR respectively, to reduce the difficulty and effort of building compilers for new processors. IR itself is not very critical. This holds true for MLIR, too. That's why LLVM and MLIR are called compiler infrastructures.

We agree with the reviewer that the key contribution of LLVM is its ability to optimize and convert between IRs. Before the neuromorphic community can optimize the conversion of microcode, the neuromorphic community needs to land on what operations that microcode describes. Although the specific form of IR may not be critical, the formalism can provide an additional method for enabling linking between the various layers of software for different hardware systems. It is also our view that the first step towards a general compilation infrastructure can be enabled efficiently by using intermediate representations. After the first step, an initial common representation, such as NIR, can provide indicators of where the system could be tuned for different applications.

We do not claim that NIR advances or replaces compiler infrastructure; NIR is a separate tool to MLIR/LLVM with a different goal to link between heterogeneous components of

application-specific computing systems. The table in Appendix B is meant to give the reader context on how it fits alongside these tools.

As the Reviewer has indicated, building optimized conversion codes that are built upon inspiration from MLIR is a logical next step.

Author Action

- We have added the goal of building optimized conversion codes to the roadmap of NIR and welcome open-source contributions to enable future compiler infrastructure-like benefits in the continuous-time domain
- We highlight that we are **not** an alternative to MLIR/LLVM by including a comment in Table B:
 - *“To clarify the relation of NIR to other compilers, code generators, optimizers, and network modelling tools, we provide the following comparison table. Most of the frameworks in the table solve different problems. As such, the table provides high-level and independent descriptions of each framework to situate NIR in the literature.”*
- Added the following to the related work section:
 - *“Compiler infrastructures like LLVM, and later MLIR, has been leading the field in optimizing and transforming code for a wide range of programs and architectures.”*

2. Authors seem to think that current compilation infrastructures such as MLIR cannot represent continuous-time dynamical systems. I disagree with that. The root of MLIR's design advantage lies not in its existing multiple dialects, but in the dialect mechanism itself, which can be flexibly extended and descended layer by layer. NIR may serve as a high-level dialect in MLIR. These two can't be compared directly, and as mentioned above, the contribution of NIR has nothing to do with the key of MLIR.

We agree with the reviewer that MLIR can represent continuous-time dynamical systems. Namely, by representing dynamical systems primitives, similar to what we do in NIR. The “circuit” project is one such example (<https://github.com/llvm/circuit>). From that perspective, MLIR can, in fact, be useful for an intermediate representation like NIR, which we also pointed out in our previous letter and in the paper when we discuss future work.

When it comes to evaluating the continuous-time dynamical system, however, it is our understanding that MLIR is based on the polyhedral model where iterations and loops are viewed as points on a lattice. Although the polyhedral model lends itself well to various transformations and optimizations, the quantized instructions in the lattice are not continuous. So, it would *not* be feasible to map from the quantized lattice to any arbitrary continuous representations without approximations, as in analog hardware.

Author Action

- We added a passage in Section 1.1 related works section to highlight that MLIR is capable of representing continuous-time dynamical systems, but it does so using discrete nodes on a lattice that would require some level of extrapolation to reconstruct as a continuous-time graph in continuous-time hardware.
- We added a passage in Section 2.4 relation between NIR and existing intermediate representations stating that LLVM and MLIR can represent the same primitives, but are designed to optimize discrete instructions.

3. Does this in itself exacerbate fragmentation? This can be achieved with some extensions to existing IR or programming languages.

We chose to implement NIR as a stand-alone IR for simplicity and clarity. It was a logical first step towards a linking between continuous-time and discrete representations that was feasible to develop between many academic and industrial collaborators.

The numerical discrepancies that we have pointed out in our work have not been specifically highlighted before with existing tools to our knowledge, which we argue already demonstrates the value of NIR. The new section 2.4 “Relation between NIR and existing intermediate representations” describes how we believe NIR sets itself apart from other IRs. One important point is that we do not aim to replace existing software approaches, but rather serialize and integrate them as seamlessly as possible. For example, the SpiNNaker2 backend relies on PyNN to convert NIR graphs, showing that the two representations are not mutually exclusive. We hope that end-users eventually can be perfectly oblivious to NIR and use a familiar frontend to describe their model, while NIR implicitly converts their model to a hardware platform. We expect this to contribute to less fragmentation while aiding wider applications.

Reviewer #2 (Remarks to the Author):

Thank you for addressing my review.

In general, I believe the field would benefit from moving away from this artificial frontier of those mathematical abstractions. The important and hard question for the field is whether any of these models and hardware are better than other systems (deep learning with custom accelerators). This theoretical question is not solved by the model presented here, and there is the danger that the efforts put into the homogenization of the field around models which under-perform on benchmarks (in comparison to machine learning models) would slow the field down rather than bringing us forward toward successful neuromorphic mathematical models.

Our work is not focussed only on abstractions, but also on demonstrating the advantages of specific representations for different applications. It is our hope that our work shows both the advantages and the potential applications to different systems, by choosing appropriate intermediate representations.

- **R2: “there is the danger that the efforts put into the homogenization of the field around models which underperform would slow down the field.”**

We understand the concern, though we argue that NIR will instead speed up the findings of the neuromorphic community, whether they are positive or negative results. One of the key aspects of our work is to enable the ability to use representations linking applications to hardware systems. This is not intended to homogenize the field, but to examine how computing can be made efficient across heterogeneous systems, even within the domain of neuro-inspired computing.

The continuous-time primitives in NIR are already in heavy use across software libraries and neuromorphic accelerators. By providing a way to benchmark models across hardware platforms, this gives definitive and clear quantitative results on the value of a given model. NIR allows researchers to come to this conclusion much faster than they otherwise would, independent of whether such models are beneficial or suboptimal.

Furthermore, NIR provides an option for neuromorphic researchers to port models across hardware backends, and does not exclude the use of other primitives: as mentioned in our response to Reviewer 1, the SpiNNaker2 backend relies on PyNN to convert NIR graphs, showing that NIR can work with other representations, too.

Ultimately, NIR promotes reproducibility, and can enable rapid iterations on experiments across various hardware backends. By demonstrating that we can (1) model disparate neuromorphic models in pre-existing software frameworks, (2) map these to heterogeneous hardware systems, and (3) provide systematic methods for comparisons, we argue that NIR will catalyze further research in representations and more complex applications.

- **R2: “I believe the field would benefit from moving away from this artificial frontier of those mathematical abstractions.”**

The mathematical abstractions that we describe are not for the sake of theoretical insight alone, but rather for enabling neuromorphic engineers with applications to move across platforms more efficiently, ultimately expanding the possible realm of applications.

On the other hand, there is an important need to compare experimental hardware. The interoperable software being developed here might be a crucial milestone towards a systematic comparison of recent neuromorphic hardware which was designed for the previous generation of neuromorphic models. In this sense, the work is important to enable a careful comparison of contemporary simulation substrates with a fixed model. Unfortunately, the most valuable comparison results are not fully studied here (resource efficiency, scaling capability).

Resource Efficiency and Scaling Capability are critical factors. However, the intent of this research paper is mainly to demonstrate the potential of using intermediate representations to study and analyze multiple hardware systems and applications. As a result, we expect this to

initiate several research threads, including the study of resource efficiency and scaling capability.

For example, it took significant effort to elicit the level of abstraction and specification for the graph structure in the paper. This is, in our opinion, an essential effort that bridges science with engineering, including highlighting essential differences between the models and platforms. We have included a separate project in our roadmap that aims to measure energy utilization in a fair and transparent manner, hosted by Professor Sadasivan Shankar at Stanford.

In addition, we believe that we cannot meaningfully compare such metrics before we have a common understanding of *what* we compare. In fact, this has been one of the main motivations for developing NIR, as it targets interoperability rather than hardware benchmarking. But in enabling interoperability, it makes hardware benchmarking more systematic for research.

In discussing resource measurements with neuromorphic hardware vendors, we realized that transparently and fairly showing the footprints of the different hardware is a significant undertaking in of itself. For example, Intel Loihi and SpiNNaker enable the use of custom microcoded neurons and usage in a server environment. This flexibility is at the cost of efficiency, and any metrics we present in the paper would be biased towards less flexible systems, such as the SynSense accelerators. A table comparison of energy usage could relay a misleading message to readers, and expand the scope of the paper. As Reviewer 1 pointed out, 'a good IR [should] be independent of the underlying hardware implementation.'

In addition, NIR can represent models of arbitrary scale, limited only by the underlying hardware of (i) the system that converts a model to and from the NIR format, and (ii) the neuromorphic hardware that runs such a model. At present, the largest scale neuromorphic hardware that can run NIR compatible models is Hala Point, consisting of 1.15 billion neurons distributed across over 1k Loihi 2 processors.

Author action:

- We added considerations for scale in the discussion to clarify how NIR relates to the scaling properties of the neuromorphic hardware systems, with specific reference to the largest-scale neuromorphic systems available (i.e., Hala Point).
- We have included the distinct and separate project of neuromorphic hardware benchmarking in the project roadmap

My conclusion is that, although the work is worth being published as a step toward a longer-term achievement. I am doubtful that the revised manuscript will be impactful as such.

Although we appreciate this input, we respectfully disagree. We would like to add that the preprint has already been cited eight times by external authors since November 2023. In addition, the paper has been featured in several workshops, and derivative works are being actively developed (cf. our roadmap at <https://neuroir.org/docs/roadmap.html>) including in SLAC National Laboratory/Stanford as indicated above. The reception and support from the

field has been genuinely encouraging, and we hope that multiple research ideas will be further explored in the future. We also look forward to accommodating and continuing to collaborate with the community.

Reviewer #3 (Remarks to the Author):

I would like to thank the authors for carefully responding to and addressing my comments/concerns. In my opinion, the paper is now suitable for publication in Nature Communications. I would also like to thank the authors for their ambitious vision and clearly documenting and making their codes available - developing open source and community orientated tools is often a thankless effort.

We would like to again thank the reviewer for their thoughtful and relevant comments and insights. We will continue to strive for openness and reproducibility, revising NIR and potential derivative works to accommodate a growing and, we hope, more rigorous scientific field.

REVIEWERS' COMMENTS

Reviewer #1 (Remarks to the Author):

The authors have revised or added to some of the statements about their contributions to make the positioning and impact of this work more precise.

My conclusion is that, the work is worth being published as a step toward a longer-term achievement.

Reviewer #2 (Remarks to the Author):

I would like to thank the authors for addressing my comments. I have raised my concerns regarding the risks and slowing down the progress of the field, and regarding the positive impact of this paper. The last response from the author highlights our diverging opinions.

On the technical and scientific side, the papers content is correct and may be published as it is. The aspects which are not consensual are on a high level, I am likely wrong about them and further comments are unlikely to make improvements in the paper at the point.